# When is Unsupervised Disentanglement Possible?

**Daniella Horan and Eitan Richardson and Yair Weiss**
School of Computer Science and Engineering
The Hebrew University of Jerusalem
Jerusalem, Israel
{daniella.horan,eitan.richardson,yair.weiss}@mail.huji.ac.il

## Abstract

A common assumption in many domains is that high dimensional data are a smooth nonlinear function of a small number of independent factors. When is it possible to recover the factors from unlabeled data? In the context of deep models this problem is called "disentanglement" and was recently shown to be impossible without additional strong assumptions [17, 19]. In this paper, we show that the assumption of local isometry together with non-Gaussianity of the factors, is sufficient to provably recover disentangled representations from data. We leverage recent advances in deep generative models to construct manifolds of highly realistic images for which the ground truth latent representation is known, and test whether modern and classical methods succeed in recovering the latent factors. For many different manifolds, we find that a spectral method that explicitly optimizes local isometry and non-Gaussianity consistently finds the correct latent factors, while baseline deep autoencoders do not. We propose how to encourage deep autoencoders to find encodings that satisfy local isometry and show that this helps them discover disentangled representations. Overall, our results suggest that in some realistic settings, unsupervised disentanglement is provably possible, without any domain-specific assumptions.

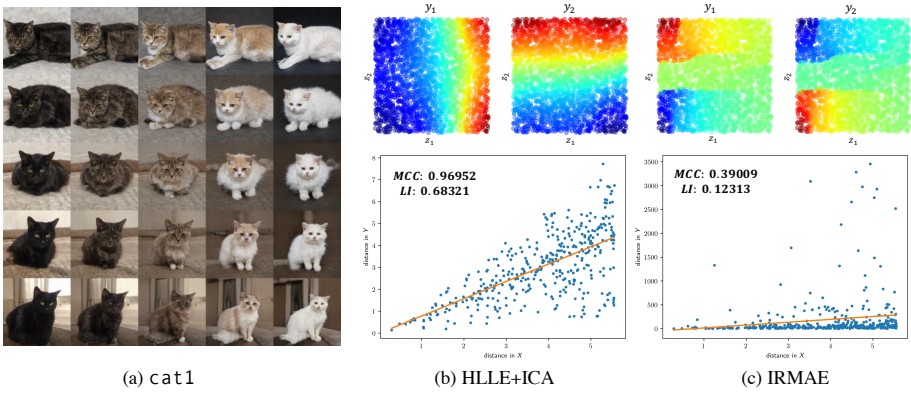

(a) cat1        (b) HLLE+ICA        (c) IRMAE

Figure 1: (a) Manifold of highly realistic images with two independent latent factors. Given unlabled images, is it possible to recover the true factors? HLLE+ICA (b) succeeds while a baseline autoencoder (c) learns a highly entangled embedding. Similarly, the representation found by HLLE+ICA approximately satisfies local isometry while the encoding of the baseline autoencoder does not (bottom). In this paper we show that the assumptions of local isometry and non-Gaussianity are sufficient to provably recover disentangled representations.

35th Conference on Neural Information Processing Systems (NeurIPS 2021).

# 1 Introduction

With the recent success of deep learning approaches, there has been a resurgence of interest in the problem of recovering the independent factors that generate high dimensional data. As a motivating example, consider the `cat1` manifold shown in figure 1a. Each image is of size $256 \times 256 \times 3$, but is generated as a smooth, nonlinear function of just two independent latent factors: the fur color and pose of the cat. Is it possible to recover these latent factors, given only samples of images?

This problem is closely related, yet different from the problem of learning a deep generative model from data. Given a training set $\{x^i\}$, deep generative models find parameters $\theta$ such that $x^i = g_\theta(z^i)$ where each latent variable $z^i$ is sampled from a base distribution, and the different components of $z$ are independent. Tremendous progress in deep generative models has been achieved by methods including Generative Adversarial Networks (GAN) [9, 1, 16], Variational Auto-Encoders (VAE) and their variants [18] and optimization-based approaches [4].

While deep generative models have achieved remarkable success in generating realistic images given a random input, the latent variables that they find are often very different from the desired "independent factors" [19]. Returning to our motivating example in figure 1a, a deep generative model may generate perfectly realistic images given random latent vectors $z$, where some of the coordinates of $z$ have only a weak relationship to the true latent factors, and no single coordinate encodes a latent factor perfectly, as desired.

Figure 1 illustrates this issue. In the top of column (c) we show the representation learned by a particular autoencoder [15] for the data on the left (see figure 5 for additional baseline autoencoders). The embedding is shown as a function of the ground truth factors ($z_1$: fur color, $z_2$: cat pose). The autoencoder learned two latent dimensions, $y_1$ and $y_2$, and we plot $y_1(z_1, z_2)$, $y_2(z_1, z_2)$ using the "jet" colormap that goes from blue to red. If each learned dimension encodes only one factor, then these plots should have contours that are parallel to the axes. For this particular autoencoder, however, this is not the case. Even though it gives very low reconstruction loss (see supplementary), it can be seen that neither fur color nor pose are directly encoded by either of the learned factors. Such a representation is called "entangled" [3], and there has been much recent work attempting to quantify the extent to which a learned representation is "disentangled" [8, 11, 5].

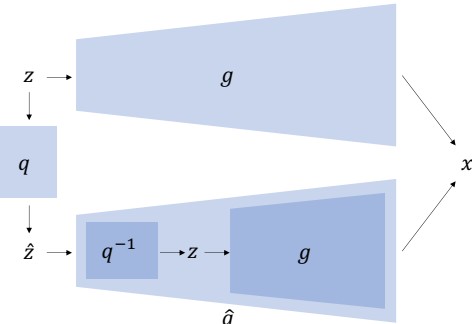

Figure 2: For any generative model, $g$, we can always apply a transformation $q$ to the latent code $z$ and add $q^{-1}$ to the generator to obtain an equivalent generative model, $\hat{g}$. In the linear case, if the code $z$ is sufficiently non-Gaussian any linear transformation $q$ will make the distribution more Gaussian and so it is possible to differentiate between the true $z, G$ and any alternative $\hat{z}, \hat{g}$. In the non-linear case, even if we assume a non-Gaussian prior, there exist many invertible functions $q$ that preserve the distribution. This enables the construction of $\hat{g}$, such that the pair $\hat{z}, \hat{g}$ is indistinguishable from the true $z, g$.

An intriguing result from the recent literature on disentanglement is that the problem is impossible without additional assumptions, even with infinite data [19, 17]. This issue is well-understood in the context of linear functions of latent factors, i.e. Independent Component Analysis [12]. In the linear ICA setting, it is known that while the problem is non-identifiable when the latent factors

are Gaussian, it can be solved if the latent factors have a sufficiently non-Gaussian distribution. The recent work of [19, 17], points out that in the nonlinear case, the identifiability problem is much more severe: even if the independent factors are non-Gaussian, there can be many non-linear transformations $\hat{z} = q(z)$ that preserve the distribution of $z$. By defining $\hat{g}(\hat{z}) = g(q^{-1}(\hat{z}))$, for $g(z) = x$, we can perfectly generate the data from latent variables that have the desired distribution yet are highly entangled (see figure 2 for a graphical representation of the impossibility result). This suggests that disentanglement is impossible without additional strong assumptions on the architecture and the data even with infinite data, and indeed Locatello et al. demonstrated empirically that many deep autoencoder methods fail to learn a disentangled representation.

In this paper we show the power of one very general assumption that can enable the learning of disentangled representations. The assumption of *local isometry*, i.e. that an $\epsilon$ change in the latent representation corresponds to an $\epsilon$ change in the observed variables. In the bottom of figure 1b,c we show a scatter plot of the distances $d(x^i, x^j)$ and $d(y^i, y^j)$ for randomly chosen pairs of nearby points. If the embedding satisfies local isometry, these points should lie on a straight line. As can be seen, the highly entangled autoencoder embedding does not satisfy local isometry: there is almost no correlation between the magnitude of change in the embedding space $Y$ and the magnitude of change in the original input space $X$.

Specifically, we prove that as the number of datapoints goes to infinity, a combination of two classical methods: Hessian Eigenmaps [7] and fastICA [12], is guaranteed to find a disentangled representation when the manifold parameterization satisfies local isometry and the base distribution is sufficiently non-Gaussian. Figure 1b shows the result of our proposed method on the same data that the autoencoder was trained on. The contours in the plots are now axis-aligned, indicating that a perfectly disentangled representation was learned. As shown in the bottom plot, there is a strong correlation between the pairwise distances of points in the input space $X$ and their corresponding representations in the embedding space $Y$.

In the remaining sections, we first derive analytical conditions on manifold parameterizations that guarantee that disentanglement will be possible; and discuss settings in which these conditions hold. We leverage recent advances in deep generative models to construct manifolds of highly realistic images for which the ground truth latent representation is known, and test whether different methods succeed in recovering a disentangled representation. For many different manifolds, we find that methods that explicitly optimize local isometry and non-Gaussianity consistently find the correct latent factors, while baseline deep autoencoders do not. Finally, we discuss ways to encourage deep autoencoders to find encodings that satisfy local isometry, and show that this helps them discover disentangled representations. Overall, our results suggest that in some realistic settings, disentanglement is possible without any domain-specific assumptions.

## 2   Local Isometry + non-Gaussianity make disentanglement possible

Our main theoretical result shows that a combination of two classical algorithms is guaranteed to find disentangled representations under two classical assumptions: local isometry and non-Gaussianity. We first define the assumptions and the algorithms and then state the result.

**Local Isometry:** Intuitively, a mapping from a latent representation $z$ to an observed vector $x$ satisfies local isometry if an $\epsilon$ change in $z$ corresponds to an $\epsilon$ change in $x$. Formally, a mapping $g$ from $z$ to $x$ is said to satisfy *local isometry* if for all $z$, the Jacobian $J(z) = \frac{\partial g}{\partial z}$ satisfies:

$$J(z)^T J(z) = I$$

As we discuss in section 3, local isometry has been widely used in manifold learning algorithms and any manifold can be approximated to arbitrarily high accuracy using an isometric parameterization.

**Non-Gaussianity:** Given a distribution over $p(z)$, we say that $z$ is sufficiently non-Gaussian, if a linear ICA algorithm is guaranteed to recover $z$ (up to permutation) from a linear mixing $Az$, given infinite training examples. As mentioned above, this assumption is crucial to enable disentanglement in linear generative models.

The two algorithms that we will use are HLLE [7] and fastICA [13].

**Hessian Eigenmaps (HLLE)** belongs to a family of manifold learning techniques that first estimate a local parameterization around each datapoint and then align the local parameterizations by solving

an eigenvector problem (e.g. [25, 30, 28]). Specifically, it is based on the insight that the local isometry assumption means that local PCA will recover a parameterization that is locally isometric to the true parameterization. Given a dataset $\{x^i\}_{i=1}^M$, an integer $K$ that determines the size of the neighborhoods, and $d$ the dimensionality of the manifold (or equivalently the dimension of the latent code), the algorithm proceeds in the following steps:

- For each point $x^i$, identify its $K$ nearest neighbors using Euclidean distance.

- Perform a local principal components analysis (PCA) within each neighborhood and use the principal components in each neighborhood to define a global, $M \times M$, matrix $H$.

- Calculate the $d+1$ eigenvectors of $H$ with smallest eigenvalues. Discard the trivial, constant eigenvector to obtain a $d$ dimensional embedding $Y$. The embedding of $x^i$, which is a vector of length $d$ denoted $y^i$ is then computed from the $i$th index of the $d$ eigenvectors.

As shown in [7], given infinite samples from a manifold for which the mapping from $z$ to $x$ satisfies local isometry, the HLLE algorithm is guaranteed to find a representation of $x$ that is the same as $z$ up to a linear transformation: $y^i = Az^i + b$ for some matrix $A$.

The **fastICA** algorithm [12] searches for projections of the data $(u = w^T x)$ which maximize the deviation between the expectation of a nonlinear function of the projection, $G(u)$, and the same expectation with a Gaussian distribution. It is guaranteed to be consistent "for most reasonable choices of $G$, and distributions of the sources. In particular, if $G(u) = u^4$, the condition is fulfilled for any distribution of non-zero kurtosis. In that case, it can also be proven that there are no spurious optima"[13]. In other words, if $p(z)$ has nonzero kurtosis and we observe infinite samples from $x = Az$, then fastICA with $G(u) = u^4$ is guaranteed to correctly recover $A$.

Finally, we refer to HLLE+ICA as an algorithm that first runs HLLE and then runs ICA on the embedding found by HLLE. We are now ready to state our main result.

**Definition:** A representation $y(x)$ is said to be a disentangled representation of the true latent factors $z$ if there exists a permutation $\pi$ and a pointwise nonlinearity $f$ such that $y = \pi(f(z))$.

**Theorem 1:** Let $\{x^i\}$ be samples from a manifold $x = g(z)$. Assume that the support of $p(z)$ is connected. If there exists an alternative representation of the manifold $x = g_2(s)$ with $s = Af(z)$, for any matrix $A$ and pointwise nonlinearity $f$, such that $g_2$ satisfies local isometry and $f(z)$ is sufficiently non-Gaussian; then as the number of samples goes to infinity, HLLE+ICA will provably find a disentangled representation of the true latent factors $z$.

**Proof:** Donoho and Grimes [7] showed that as the number of examples goes to infinity, the $H$ matrix constructed in the HLLE algorithm converges to the (continuous) Hessian operator on the manifold whose $d+1$ eigenfunctions with zero eigenvalue are spanned by the latent factors. They also showed that if the support of $p(z)$ is connected, then HLLE will recover the isometric embedding of the manifold, up to a linear transformation. In our setting, the results of Donoho and Grimes guarantee that HLLE will recover $s$ up to a linear transformation. Since $s$ is assumed to be a linear mixing of independent factors, the assumption of non-Gaussianity means that running fastICA on the output of HLLE will provably recover a disentangled representation.

Figure 3 illustrates our theorem for a two dimensional manifold created by translating white squares on a black background. The location of each square is sampled from a uniform distribution. The mapping from $z$ (the horizontal and vertical position) to $x$ (the image of the square) has been shown to satisfy local isometry by [6]. Thus, HLLE (left) is guaranteed to find a representaton that is related to the true positions by a linear transformation. However, this is not a perfectly disentangled representation just yet. Nevertheless, since the distribution is uniform, not Gaussian, applying ICA on the result of HLLE (middle) indeed recovers a perfectly disentangled representation.

Note that the definition of local isometry is sensitive to rescalings: if we parameterize the squares manifold in terms of horizontal and vertical position, then it will satisfy local isometry. However, we can also parameterize the same manifold using $\log(x), \exp(y)$ and such a parameterization does not satisfy local isometry. Nevertheless, Theorem 1 guarantees that we will still recover a disentangled representation, regardless of the rescaling of the true factors. In the supplementary material, we show that another classical manifold learning algorithm, Laplacian EigenMaps (LEM), followed by fastICA, will also recover disentangled representations under certain conditions, but unlike HLLE+ICA, LEM+ICA is very sensitive to rescalings.

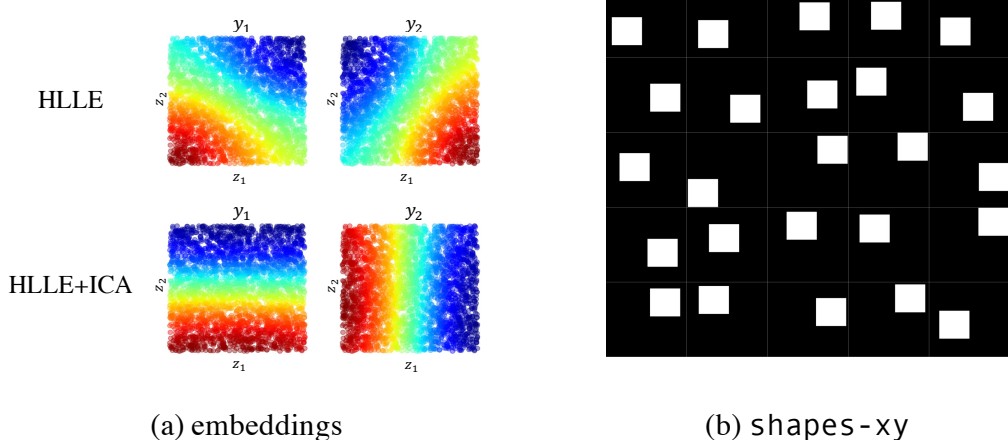

(a) embeddings            (b) `shapes-xy`

Figure 3: The results of running HLLE+ICA on 3000 samples from a 2D manifold created by translating a white square on a black background. The results of [6] guarantee that this manifold satisfies local isometry and our theorem guarantees that HLLE followed by ICA will recover a perfectly disentangled representation. As can be seen, the algorithm work perfectly even with finite data.

## 3 Are the assumptions reasonable?

Our theorem is based on two assumptions: non-Gaussianity and local isometry. While both assumptions have been used in machine learning for over 20 years, it is natural to ask whether these assumptions are reasonable in the context of disentanglement.

Regarding non-Gaussianity, this assumption has been successfully used in linear ICA in a wide range of real-world applications including analysis of gene expression data, extracting features from stock market data and removing artifacts from EEG data [14]. In particular, linear ICA algorithms can successfully disentangle sources with different distributions and in the presence of outliers.

The history of local isometry in manifold learning goes back to to work of Roweis and Saul [25], who used the assumption that manifolds are locally linear as the basis of LLE, and Tenenbaum et al. [29] who used the stronger assumption of global isometry as the basis of ISOMAP. Donoho and Grimes [7] showed how to interpret LLE in terms of local isometry and presented HLLE as a more principled alternative to LLE. They also showed that local isometry is a more general assumption than the global one used by ISOMAP, and which covers many interesting articulation manifolds. Taken together, these algorithms have been used in a remarkably wide range of applications.

Manifold learning approaches based on local isometry have been shown to be successful on precisely the type of manifolds that are typically used in the disentanglement literature. Many recent papers use dsprites [23], or some variant of it, which consists of simple 2D objects translating and rotating in the image plane. These types of manifolds were shown to satisfy local isometry by Donoho and Grimes [6]. An additional example that is very common in the disentanglement literature is images of faces at different poses, illumination and expressions. Donoho and Grimes [6] have shown that a manifold of images of faces with different expressions satisfies local isometry, and such datasets are successfully handled with HLLE.

As we mentioned above, any manifold can be approximated by one that satisfies local isometry. This follows from the Nash-Kuiper theorem (1956) [27] which shows that any manifold of dimension $d$ embedded in an $n$ dimensional space ($n \geq d + 1$), can be approximated to arbitrary accuracy with an isometric parameterization.

Note that this result does not contradict Gauss's Theorema Egregium which states that Gaussian curvature is preserved under isometries and hence, for example, it is impossible to find a global, exact isometry from a partial sphere to the plane. But this impossiblity is only for global, exact

isometries. Indeed we have found that running HLLE on data sampled from a partial sphere will find an embedding that satisfies local isometry almost perfectly.

As the preceding discussion shows, there always exists an approximate parameterization of a manifold that satisfies local isometry, but for our theorem to hold, this parameterization should correspond (up to a linear mixing and rescalings) with the "ground truth" factors that we seek. In the next section we ask whether this holds in realistic image manifolds.

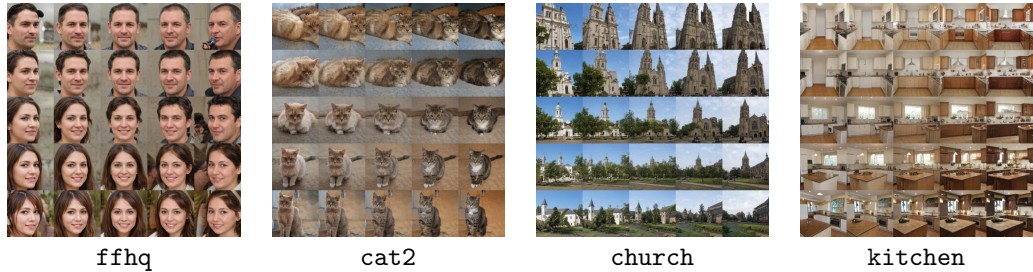

ffhq  cat2  church  kitchen

Figure 4: Realistic 2D image manifolds generated with StyleGAN and GANSpace. Each dataset consists of 3000 samples. The rows correspond to the $z_1$ direction and the columns to the $z_2$ direction.

## 4 Experiments

We conducted experiments on two simple synthetic manifolds and 8 realistic synthetic manifolds. The two simple manifolds were constructed by squares on a black background, where we varied the 2D position in the first manifold (`shapes-xy`), and the diagonal position and the color in the second (`shapes-xc`). Similar manifolds have been used in many recent papers on disentanglement (e.g. [11, 15]), and we ensured that the latent representation has a non-Gaussian distribution by sampling the latent variables from independent uniform distributions.

In addition to these simple manifolds, we also leveraged recent advances in deep generative models to create realistic image manifolds where each image is a complex, nonlinear function of a small number of known latent variables.

Specifically we use GANSpace [10] to find interpretable semantic directions in the $W$ space of StyleGAN2 [16], using PCA. To generate a $d$-dimensional manifold we select $d$ such directions and obtain their corresponding PCA components. We then uniformly sample the coefficients of these $d$ components, while fixing the coefficients of the remaining components to a constant (0). For example, for the `ffhq` manifold, we use the first two PCA components identified by GANSpace, which correspond to head pose and gender, as visible in figure 4. For the `cat2` manifold, the directions we select correspond to pose and fur pattern. The following table lists the 8 realistic manifolds that we generated using GANspace and their independent factors.

|       | ffhq   | cat1 | cat2    | horse | car       | kitchen   | bedroom      | church            |
|-------|--------|------|---------|-------|-----------|-----------|--------------|-------------------|
| $z_1$ | gender | pose | pose    | pose  | viewpoint | structure | viewpoint    | zoom level        |
| $z_2$ | pose   | color | pattern | color | color     | cabinet   | illumination | building material |

For each dataset we ran both modern deep autoencoder methods as well as the two classical methods described previously: HLLE+ICA (our proposed method) and LEM+ICA. HLLE has a single free parameter $k$ (the number of neighbors) and we find it by choosing the $k$ for which the local isometry score is maximal. The local isometry score is defined as the correlation coefficient between distances in the input space $d(x^i, x^j)$ and distances in the embedding space $d(y^i, y^j)$, when we only consider pairs of *nearby* points $x^i, x^j$ for which $d(x^i, x^j) < \epsilon$, and $\epsilon$ was set to include the closest 5% among the pairs of points. Since $d(y^i, y^j)$ is sensitive to a rescaling of the latent representations, we first compute a diagonal rescaling of the representations that minimizes the MSE between the rescaled $d(y^i, y^j)$ and $d(x^i, x^j)$. For some values of $k$ we found that fastICA did not converge (less than 5% of the runs) and we simply ignored the results with that value.

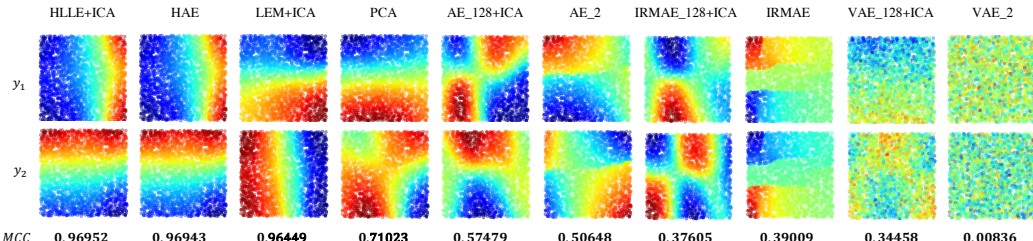

Figure 5: The embeddings found for the `cat1` manifold shown in figure 1 by various methods. None of the baseline methods manage to learn a disentangled representation.

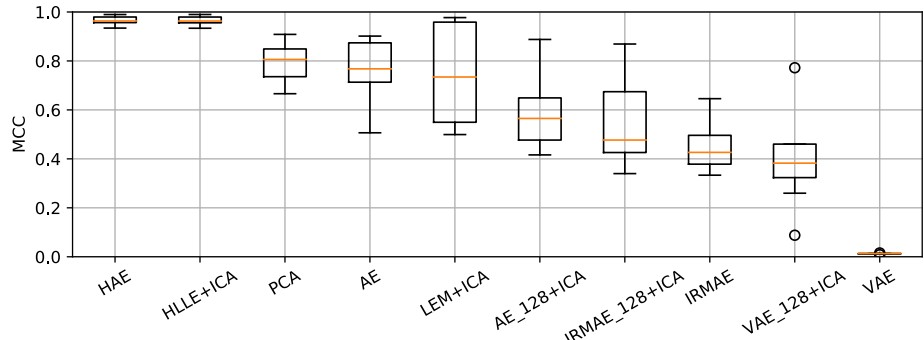

Figure 6: Boxplots for the Mean Correlation Coefficient (MCC) for all methods, over the 8 styleGAN manifolds. The box plots are computed using Matplotlib's boxplot command and the box extends from the lower to upper quartile values of the data, with a line at the median. The whiskers extend from the box to show the range of the data. Outliers are plotted as separate dots. Our proposed methods succeed in finding a disentangled representation for all 8 manifolds, while the baseline autoencoders are no better than PCA.

We also measured the performance of baseline autoencoders on the same datasets. Since these methods have many free parameters, we do not claim that the results shown here are exhaustive. Our main focus is to show that the disentanglement tasks we use here are not trivial: simply using an autoencoder with a small bottleneck along with standard regularization methods is not sufficient to ensure disentanglement, even if it excels in alternative tasks (e.g. reconstruction). The autoencoders we use were all taken from the code of a recent paper [15] and includes an implementation of a "vanilla" autoencoder (AE) with 4 conv+relu layers with stride=2 and one fully-connected layer. The Implicit Rank-Minimizing Autoencoder (IRMAE) includes in addition a number of fully connected linear layers and this has been shown to implicitly regularize the solution towards a low rank embedding. We also tested a basic implementation of a $\beta$-VAE with the value of $\beta$ that was found in [15] to provide the best performance. Results for all auto-encoders are for 100 iterations (as shown in the supplementary material, the reconstruction quality continues to improve with more iterations but the embedding is largely unchanged). Since deep autoencoders sometimes fail to learn good representations with very narrow bottlenecks, we trained two versions of all autoencoders: one in which the bottleneck was of width 2 (the true dimensionality of the manifold), and another with a bottleneck of width 128. In the second case, we ran fastICA to find a two dimensional projection of the 128 dimensional latent vector that is as non-Gaussian as possible. We also provide the results of a PCA encoder as a baseline.

Qualitative results are shown in figure 5 for the `cat1` manifold. None of the baseline autoencoder methods managed to find a disentangled representation, while HLLE+ICA and LEM+ICA did find one. Evidently the VAE suffered from "posterior collapse", i.e. "when the variational distribution closely matches the uninformative prior" [21].

As a quantitative performance metric we used the Mean Correlation Coefficient (MCC) which is standard in the nonlinear ICA community [17]. To compute it, we first solve the problem of permuting the components of the embedding $y(x)$ so it best matches the latent factors $z(x)$, using the Hungarian algorithm. We then measure the Pearson correlation between the permuted embedding and the latent variables. The results are summarized in figure 6: *our proposed method consistently found a disentangled embedding, while baseline autoencoders did not.*

As expected, run-times of the spectral methods were much faster than the autoencoders. With $k = 30$ neighbors, running HLLE took about 6 seconds per manifold and ICA took less than a second, while training the autoencoder took over an hour.

### 4.1 Hessian Auto-Encoder (HAE)

We saw that our proposed HLLE+ICA algorithm continuously succeeds in finding disentangling representations, while the autoencoders fail to do so. However, these autoencoders offer a built-in decoder and also enable out-of-sample extensions. These are two valuable advantages that HLLE+ICA lacks. In this section we propose a way to achieve both, by encouraging autoencoders to learn representations which satisfy local isometry.

In order to do so, we use a "student-teacher" paradigm. We first run HLLE+ICA and then train the autoencoder to minimize the following loss:

$$L(\theta_1, \theta_2; x, y) = \sum_i \|E(x^i; \theta_1) - y^i\|^2 + \|D(y^i; \theta_2) - x^i\|^2 \qquad (1)$$

where $\theta_1$ are the parameters of the encoder $E(x)$, $\theta_2$ are the parameters of the decoder $D(y)$ and $y$ is the embedding calculated by HLLE+ICA. This essentially decouples the learning of the encoder and the decoder. We use exactly the same encoder and decoder architectures as the baseline methods. As can be seen in the embedding visualizations in figures 7, 8, this method, which we call the Hessian Autoencoder (HAE), gives near perfect disentanglement on all manifolds, as opposed to the baseline autoencoders which often find highly entangled embeddings. This entanglement can also be seen in the interpolations in the figures. The baseline autoencoders often jump abruptly across the latent space, while HAE transitions smoothly along the manifold, gradually moving in both dimensions, as desired.

Whereas the vanilla autoencoders only strive for good reconstruction quality, the HAE also aims to learn its HLLE+ICA teacher's embeddings. As a result of its more complex loss, it typically requires more iterations to achieve the same reconstruction quality as the vanilla AE. However, with more iterations it is indeed able to achieve the same high quality results as its vanilla AE counterpart. The results in figure 7 show reconstructions with 1000 iterations for HAE and 100 iterations for the vanilla AE. A more detailed comparison can be found in the supplementary material.

## 5 Related Work

Our work was inspired by the recent work of Locatello et al. [19] who point out that "unsupervised learning of disentangled representations is fundamentally impossible without inductive biases on both the models and the data". They conducted extensive empirical experiments on "more than 12000 models covering most prominent methods and evaluation metrics in a reproducible large-scale experimental study on seven different data sets" and observed "that while the different methods successfully enforce properties encouraged by the corresponding losses, well-disentangled models seemingly cannot be identified without supervision".

We agree with the fundamental need for inductive biases, but our work suggests that a relatively general inductive bias (local isometry) makes disentanglement possible without supervision.

Khemakhem et al. [17] extend the impossibility results and relate them to classical results on identifiability of nonlinear ICA. They state that "models with any form of unconditional prior $p(z)$ are unidentifiable" and argue for using conditional priors (where $z$ is conditioned on a third variable such as time). Our work again acknowledges the non-identifiability issue without inductive bias, but shows that a relatively general inductive bias allows us to estimate the correct latent factors even with an unconditional prior.

HAE                                    AE

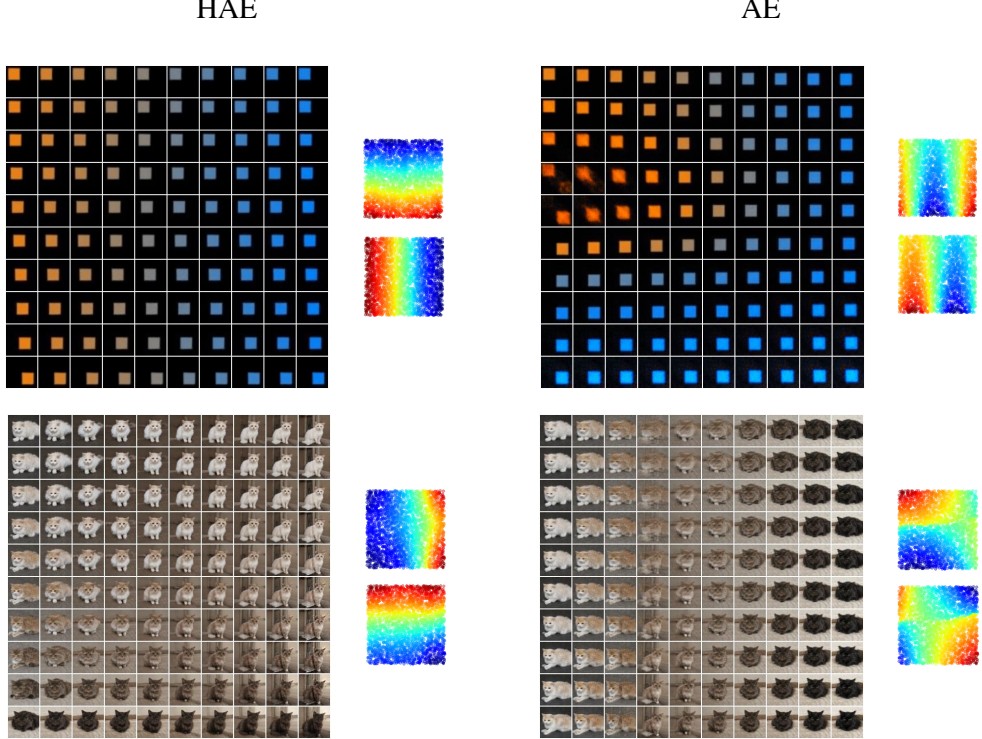

Figure 7: Interpolations in the embedding space of HAE and the vanilla autoencoder, around a chosen point (upper left image of each grid). The rows correspond to an $\epsilon$ change in the first dimension and the columns in the second. For each manifold and model, the corresponding embeddings are visualized to the right of the interpolation. On both manifolds, HAE's two directions perfectly match the true underlying factors, whereas the AE's do not. As can be seen, entangled embeddings result in discontinuous interpolations.

Atzmon et al. [2] suggested using local isometry as a regularizer in deep autoencoders and compared this regularizer to other regularizers in the context of visualization of high dimensional data. Our focus, on the other hand, has been to show that local isometry together with non-Gaussianity is sufficient to guarantee disentanglement.

Mathieu et al. [22] have argued that the stochastic encoder used by VAE may provide an inductive bias that enables disentanglement. Our experimental results, as well as those of others [20, 15], suggest that in many cases this inductive bias is not sufficiently strong to guarantee disentanglement, and our theoretical results show that local isometry is sufficiently strong.

There has been much work on performance evaluation methods for disentanglement [11, 8]. Here we used the simple MCC performance measure, and verified using visual inspection that it captures the success or failure of disentanglement. A second major difference between our work and most existing papers on evaluating disentanglement is the use of a more realistic, complex dataset that we define using StyleGAN. While we have also shown results on simple 2D shapes on constant background, we believe the StyleGAN datasets are much more realistic and capture a much wider range of latent factors that affect images.

The fact that LEM is highly sensitive to the relative scaling of different dimensions was previously pointed out by Singer and Coifman [26] and by [24]. Both of them suggested methods for improving LEM: either by pre-computing a local covariance at each point and using it do define affinity or by post-processing the LEM output. On our datasets, HLLE consistently outperformed LEM but we believe improvements to LEM can be of great interest given our theoretical results.

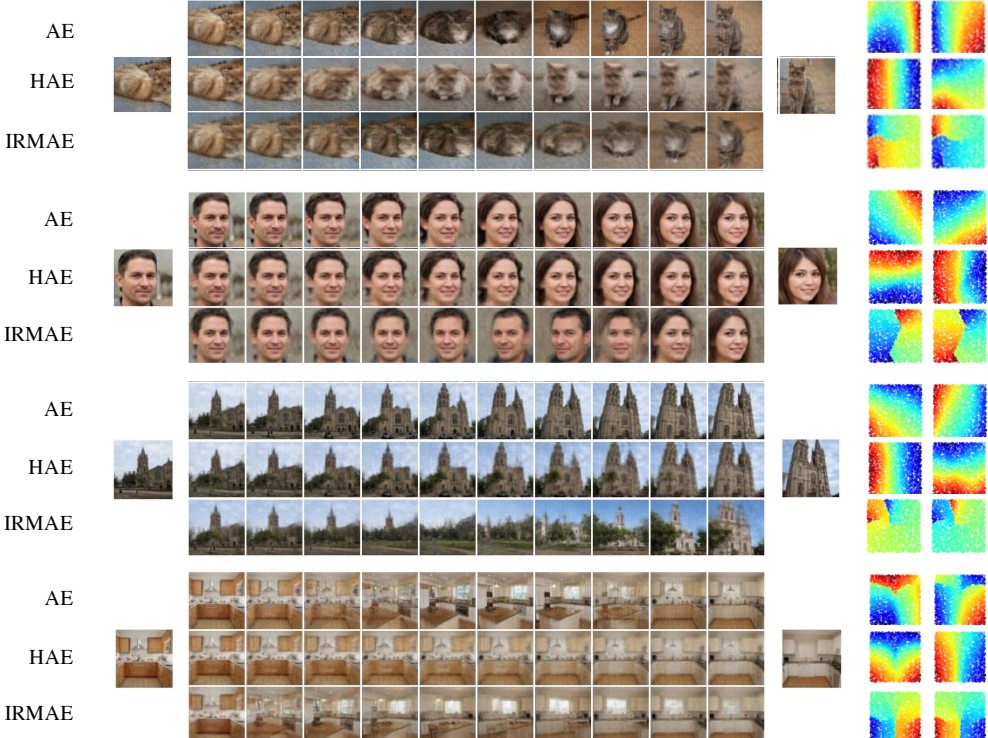

Figure 8: Interpolations in the embedding space of the different autoencoders generated by linearly interpolating between the encodings of the left and right image and then decoding. HAE transitions smoothly along the manifold between the two endpoints, while the baseline autoencoders often fail to do so. For each manifold and model, the corresponding embeddings are visualized to the right of the interpolation sequence. As can be seen, entangled embeddings result in discontinuous interpolations. All models were run for 100 iterations (see supplementary material for a discussion on the reconstruction quality).

## 6 Conclusions and Limitations

The main question addressed in our work is whether disentanglement is possible. Our theoretical and empirical results show that when we make two modest assumptions: (1) local isometry and (2) non-Gaussianity, the answer is "yes"; and a combination of two classical methods, HLLE and fastICA, can successfully learn disentangled representations in less than a minute. Our work does not contradict the recent impossiblity theorems on disentanglement which argue that the task is impossible without additional assumptions, but it shows that two very general assumptions suffice with no need for domain-specific assumptions. Moreover, in a set of experiments with highly complex realistic image manifolds, the assumptions appear to hold and disentanglement is possible with a simple, well-understood method.

While our work has shown that disentanglement is possible, we do not mean to suggest that the problem has been solved by running HLLE followed by fastICA. A major limitation of our theoretical result is that we focus on the infinite data limit and similarly the datasets we constructed correspond to a very dense sampling of the manifold (3000 images for a 2D manifold). When the sampling is sparser, it is known that spectral methods can be fooled by sampling noise. Furthermore, running spectral methods on large datasets (e.g. millions of images) is challenging and may require special-purpose approximations. We hope that our "possibility results" will encourage more research that will allow similarly successful disentanglement in a broader range of datasets.

## Acknowledgements

Support from the Israeli Science Foundation, Israeli Ministry of Science and Technology and the Gatsby Foundation is gratefully acknowledged.

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
