# When is Unsupervised Disentanglement Possible? Supplementary Material

**Daniella Horan and Eitan Richardson and Yair Weiss**
School of Computer Science and Engineering
The Hebrew University of Jerusalem
Jerusalem, Israel
{daniella.horan,eitan.richardson,yair.weiss}@mail.huji.ac.il

## 1 Laplacian Eigenmaps Disentanglement Theorem

In this section, we describe the method of Laplacian Eigenmaps [1], and derive a weaker equivalent to the theorem presented in the main paper.

### 1.1 Laplacian Eigenmaps (LEM)

Given a dataset $\{x^i\}_{i=1}^M$, a scalar $\sigma$ that determines the size of the neighborhoods, and $d$ the latent dimension, Laplacian Eigenmaps proceeds in the following steps:

- Calculate an "Affinity matrix": $W(i,j) = \exp(-\|x^i - x^j\|^2/\sigma^2)$ and use this matrix to define a Laplacian matrix $L = D - W$, where $D$ is the diagonal matrix whose diagonal elements are the sum of the rows of $W$.
- Calculate the $d$ generalized eigenvectors $Lv = \lambda D v$ with smallest eigenvalues (ignoring the trivial eigenvector which has zero eigenvalue).
- The embedding of $x^i$ is defined by the $i$th row of $V$, a matrix whose columns are the chosen generalized eigenvectors.

### 1.2 Theorem 2

**Theorem 2:** Let $\{x^i\}$ be samples from a manifold $x = g(z)$. If $g$ satisfies local isometry, $z$ is i.i.d. and each coordinate is uniform, then as the number of samples goes to infinity, HLLE+ICA and LEM+ICA will provably find a disentangled representation of the true latent factors $z$.

**Proof:** Donoho and Grimes [3] showed that as the number of examples goes to infinity, the $H$ matrix converges to the (continuous) Hessian operator on the manifold whose $d+1$ eigenfunctions with zero eigenvalue are spanned by the latent factors. They also showed that if the support of $p(z)$ is connected, then HLLE will recover the isometric embedding of the manifold up to a linear transformation. The assumption of non-Gaussianity means that running fastICA on the output of HLLE will provably recover the original independent factors. Similarly, Belkin and Niyogi [1] have shown that when the number of examples goes to infinity, the $L$ matrix converges to the (continuous) manifold Laplacian, and Singer and Coifman [7] have shown that under local isometry, the $d+1$ eigenfunctions with smallest eigenvalues are monotonic functions of the true latent functions. Since the eigenvalues are equal, LEM may recover a linear combination of the true factors and again the assumption of non-Gaussianity means that running fastICA on the output of LEM will provably recover the original independent factors.

The problem with theorem 2 is that it requires the latent factors to satisfy local isometry and the definition of local isometry is very sensitive: even when the two latent factors are pose and color, there are many different ways to parameterize each factor (e.g. pose can be measured in degrees or

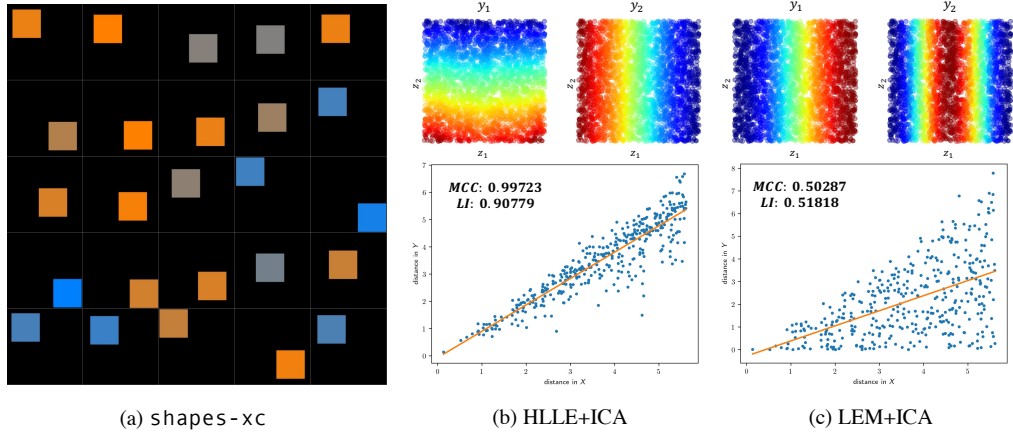

| (a) `shapes-xc` | (b) HLLE+ICA | (c) LEM+ICA |

Figure 1: An illustration of the difference between Theorem 1 and Theorem 2. (a) Samples from the `shapes-xc` manifold generated by two independent factors (diagonal position, color). The mapping from these factors to the images does not satisfy local isometry, causing LEM+ICA to fail (c), but a mapping from a monotonic transformation of the two factors to the observations does satisfy local isometry, allowing HLLE+ICA to succeed (b).

radians, color can be measured in RGB or YUV, etc.), and different parameterizations will in general not preserve local isometry. Theorem 1 (main paper) is much more general.

Figure 1 illustrates the difference between the two theorems. Images of squares are generated by two latent factors: color and location, which are sampled uniformly and independently. Changing the color of the square by $\epsilon$ has a much greater effect on the image compared to changing the location by $\epsilon$, so $f(z)$ does not satisfy local isometry and thus theorem 2 does not hold, causing LEM+ICA to fail. However, since there exists a rescaling of the latent variables that satisfies local isometry, theorem 1 does hold and HLLE+ICA succeeds.

## 2 Manifolds for which local isometry provably holds

The previous section shows that when the ground truth factors are related to an isometric encoding of the manifold, then disentanglement is possible. But this will not be the case if the "ground truth" factors are a highly entangled function of an isometric encoding. Thus referring to the first figure in the main paper, if the "ground truth" factors were the two variables found by the autoencoder rather than "fur color" and "pose", then we cannot expect HLLE+ICA to recover them. This raises the question, for what kind of manifolds and representations will the conditions of theorems 1 and 2 hold? We now give examples of seven cases.

The first five examples were given in [2] "(1) Translation of simple black objects on a white background; (2) Pivoting certain simple black objects on a white background around a fixed point; (3) Morphing of boundaries of black objects on a white background; (4) Articulation of 'fingers' of a digital 'hand'; (5) Articulation of a cartoon face by arranging its eyebrows, eyelids, and lips."[2].

Two more examples are (6) color images of aligned faces in which the latent variables control the color of the hair, eyes and mouth; and (7) gene expression data in which the latent variables activate disjoint subsets of the genes. The last two examples can be shown to satisfy the condition that $J^T J$ is diagonal (but not necessarily equal to the identity). However, we can independently reparameterize each latent factor using a monotonic transformation that makes $J^T J = I$. Thus, they satisfy the condition of theorem 1 and not theorem 2.

Figure 2 shows the results of HLLE and LEM on `shapes-xy`, a 2D manifold of squares at different positions. This is equivalent to the first case discussed in [2], and therefore the mapping from latent factors to images will provably satisfy local isometry. Together with theorem 2, disentanglement is guaranteed to be possible for this manifold. Indeed as shown in figure 2, both LEM and HLLE find

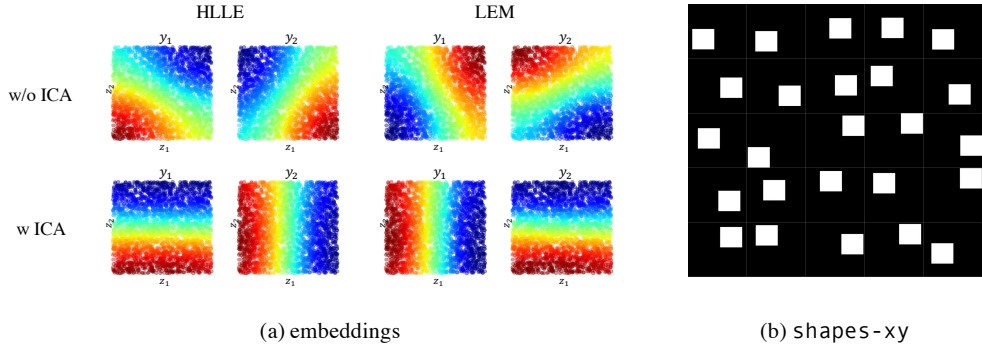

(a) embeddings                                    (b) `shapes-xy`

Figure 2: Results of running HLLE+ICA and LEM+ICA on 3000 samples from a 2D manifold created by translating a white square on a black background. The results of [2] guarantee that this manifold satisfies local isometry and our theorem thus guarantees that spectral methods followed by ICA will recover a perfectly disentangled representation. As can be seen, the algorithms work perfectly even with finite data.

the correct factors up to a linear transformation, and a perfectly disentangled representation with the addition of ICA.

Note that this list is by no means exhaustive, but is rather meant to illustrate the generality of the local isometry assumption.

# 3    Additional Experimental Results

## 3.1    Reconstructions

As described in the main paper, we based our auto-encoder experiments on the official implementation of IRMAE [5].

Figures 3, 4, 5 show reconstructions of random samples using HAE, AE, IRMAE and VAE, on the cat1, ffhq and kitchen manifolds. The VAE model did not manage to learn a 2-dimensional embedding that enables reconstruction (presumably due to "posterior collapse" as discussed in the text). HAE, AE and IRMAE models all found a 2-dimensional embedding that achieve good reconstructions, even though the three do not show similar success in terms of disentanglement. As shown in the main paper, while the HAE model finds a disentangled representation, both the AE and the IRMAE models find highly entangled ones.

Tables 1, 2 show the training loss of the trained models, on the different manifolds.

|       | ffhq    | cat1    | cat2    | horse   | car     | kitchen | bedroom | church  |
|-------|---------|---------|---------|---------|---------|---------|---------|---------|
| AE    | 0.0004  | 0.00054 | 0.00028 | 0.0002  | 0.00024 | 0.00057 | 0.0004  | 0.00045 |
| HAE   | 0.00159 | 0.00198 | 0.00099 | 0.00146 | 0.00113 | 0.00166 | 0.00068 | 0.00193 |
| IRMAE | 0.00447 | 0.0048  | 0.00259 | 0.00404 | 0.00377 | 0.00399 | 0.00252 | 0.00567 |
| VAE   | 0.64262 | 0.67661 | 0.67667 | 0.65512 | 0.50164 | 0.65761 | 0.63921 | 0.62317 |

Table 1: Training loss on the 8 realistic manifolds, at the end of training.

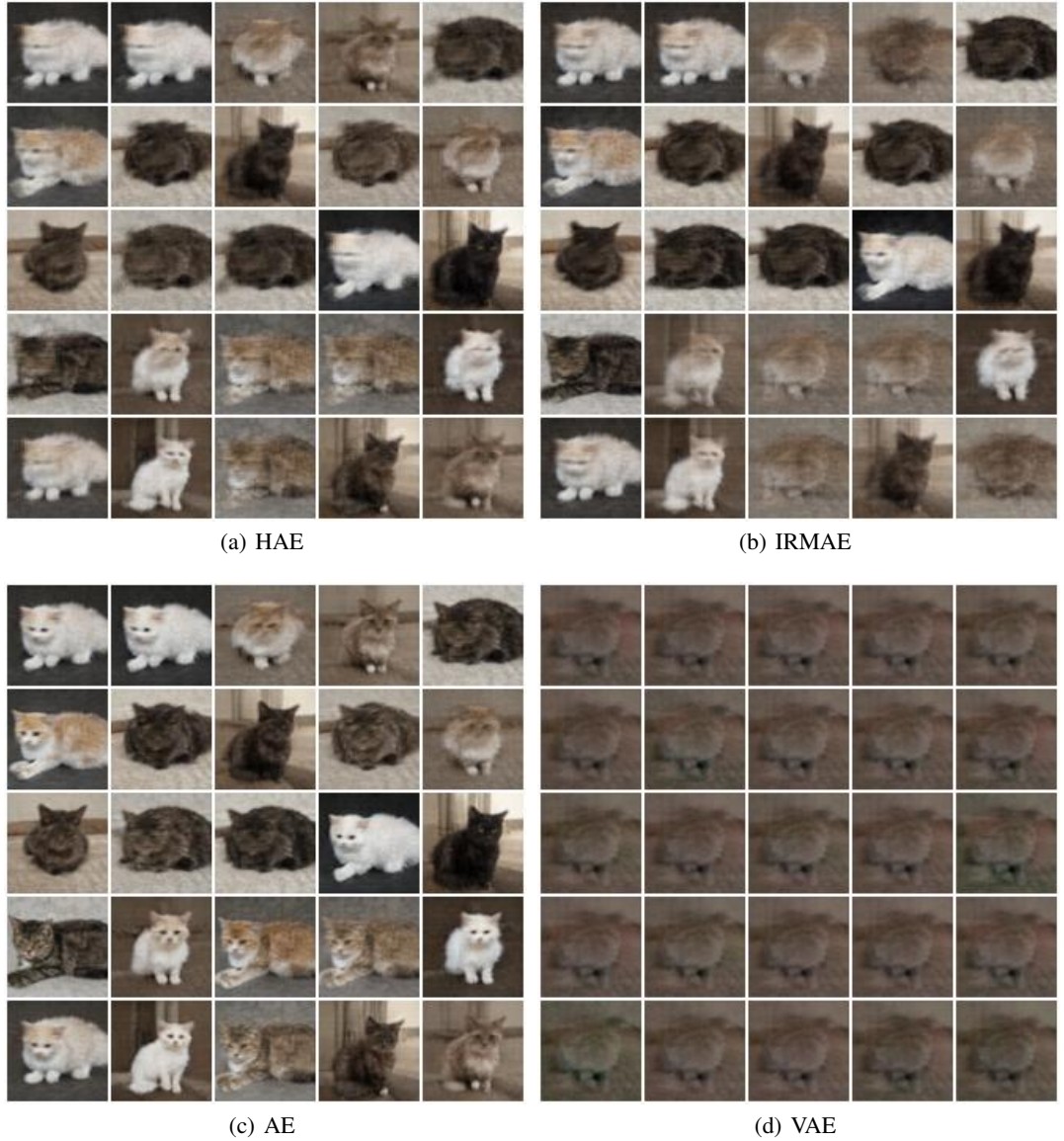

| | (a) HAE | | (b) IRMAE | |
| (c) AE | | (d) VAE | | |

Figure 3: Reconstructions of the different autoencoders, trained on `cat1`. Although our results indicate that the embedding of IRMAE (b) is more entangled than that of the HAE (a), both produce similar reconstructions.

| | shapes-xy | shapes-xc |
|---|---|---|
| AE | 0.0002 | 0.0001 |
| HAE | 0.00346 | 0.00061 |
| IRMAE | 0.01524 | 0.00363 |
| VAE | 0.2579 | 0.21693 |

Table 2: Training loss on the 2 synthetic `shapes` manifolds, at the end of training.

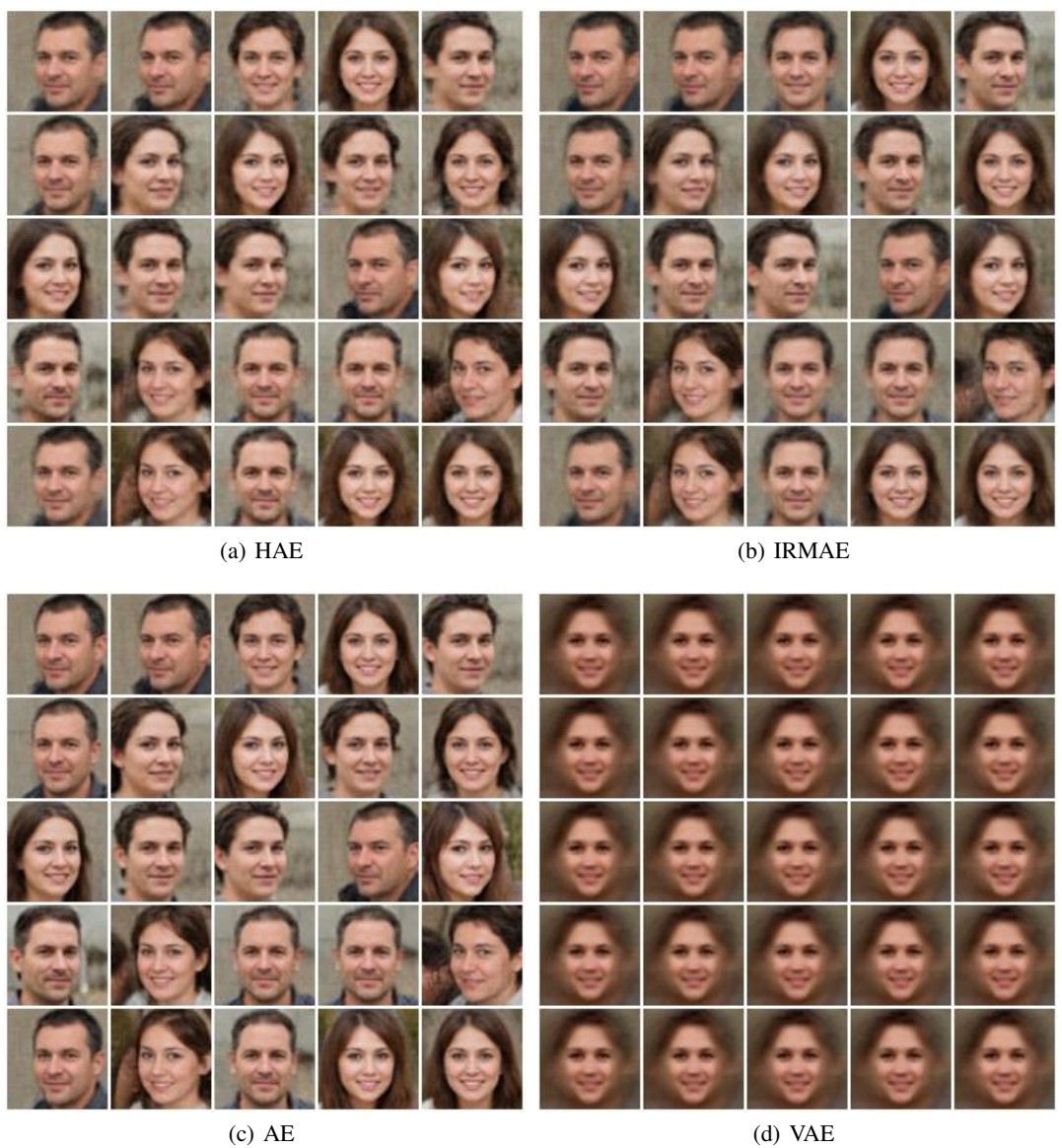

(a) HAE

(b) IRMAE

(c) AE

(d) VAE

Figure 4: Reconstructions of the different autoencoders, trained on `ffhq`.

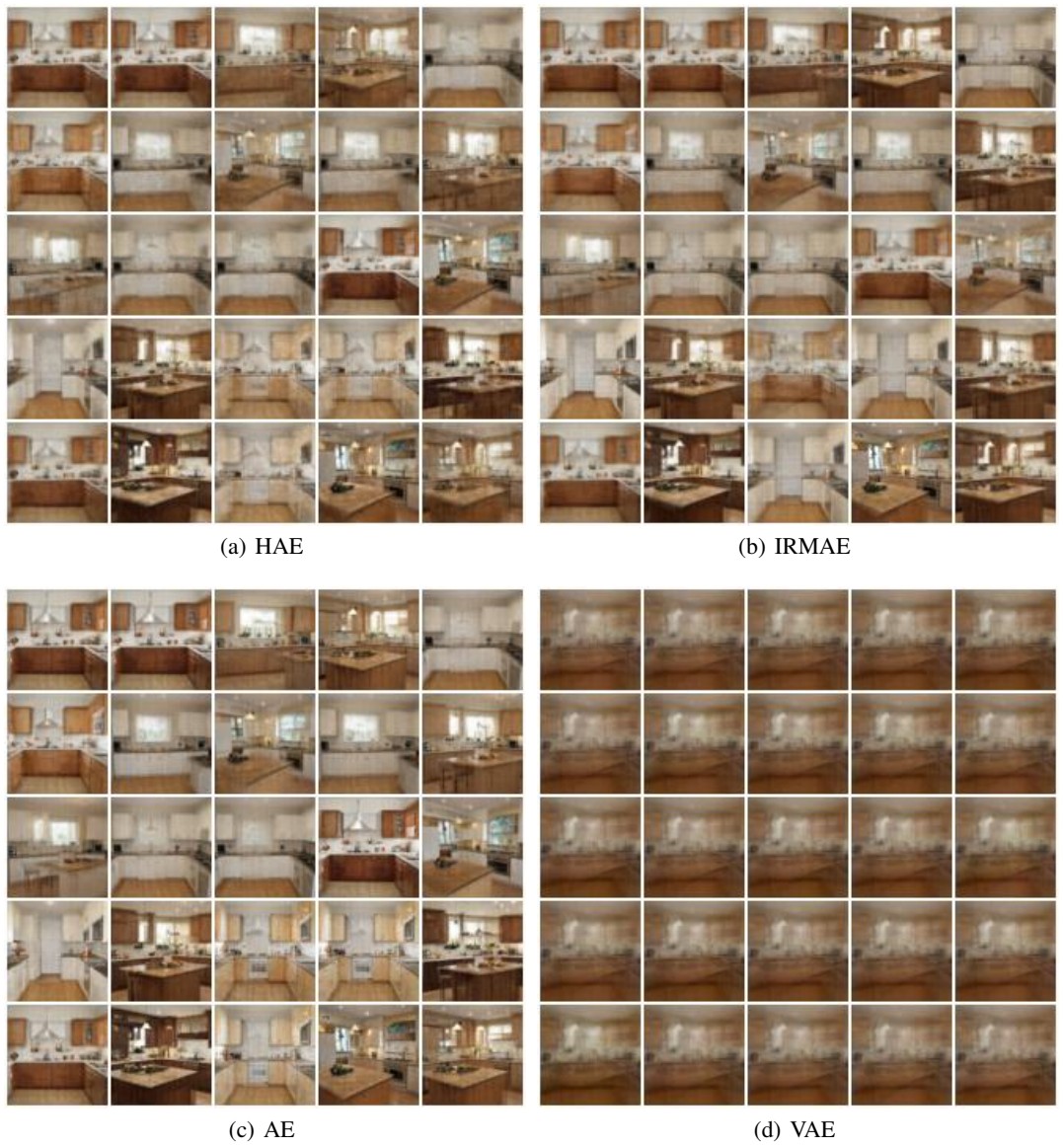

Figure 5: Reconstructions of the different autoencoders, trained on `kitchen`.

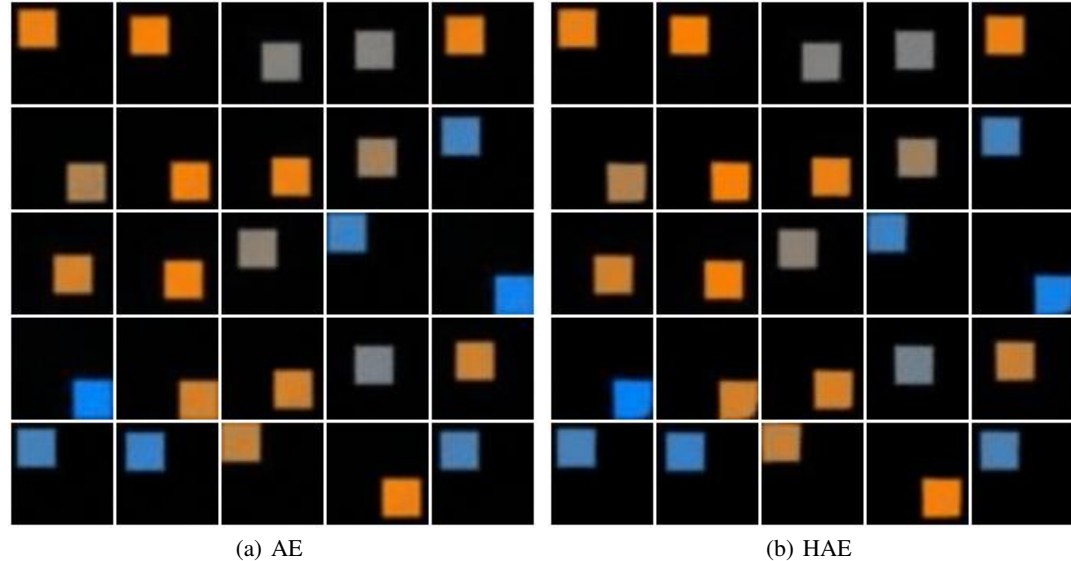

| (a) AE | (b) HAE |

Figure 6: Reconstructions of the AE and HAE models for the `shapes-xc` manifold. The AE was trained for 100 epochs, and the HAE was trained for 1000 epochs.

## 3.2 Resolution of the Generated Images

In the previous sections and in the main paper, we evaluated the different models on the tasks of reconstruction and interpolation. A visual inspection of the results suggests that the HAE model generates images that are slightly less sharp than those of the vanilla autoencoder. In this section, we investigate this discrepancy.

The HAE model has two terms in its loss, and consequently tries to achieve two goals - a disentangled latent space and good reconstructions. Therefor, it is not surprising that it may take longer for it to achieve similar results to the vanilla AE, which only aims to minimize a single reconstruction loss. The question is whether it is able to do so when we allow for longer training, or is it inherently incapable of generating high resolution images, regardless of the number of epochs.

Figures 6, 7, 8, 9 show two sets of images reconstructed by HAE and by the vanilla AE, with different numbers of epochs. We observe that in order to generate similar high resolution images, HAE indeed requires more epochs, compared to AE. Nevertheless, and more importantly, it is clear that when trained for longer, HAE is in fact able to achieve similar results.

Figures 10, 11 show the losses of the two models, for different numbers of epochs, on the `cat1` and `shapes-xc` manifolds. As can be seen the vanilla AE achieves slightly lower values than the HAE model. However, a closer examination of the more advanced stages of training indicates this gap decreases with time.

## 4 Realistic 2D Manifolds

In this section we provide additional information about our proposed method for generating 2D manifolds of realistic high-dimensional natural images. The eight manifolds we generated are shown in figures 12, 13 and random samples from these manifolds, in figure 14.

**The generative model.** We base our manifold generation on StyleGAN2 [6] since it successfully models the distribution of several natural image datasets, from which it is able to generate high quality novel samples. In addition, the Perceptual Path Length (PPL) regularization term employed by StyleGAN2 improves the smoothness of the generated manifold.

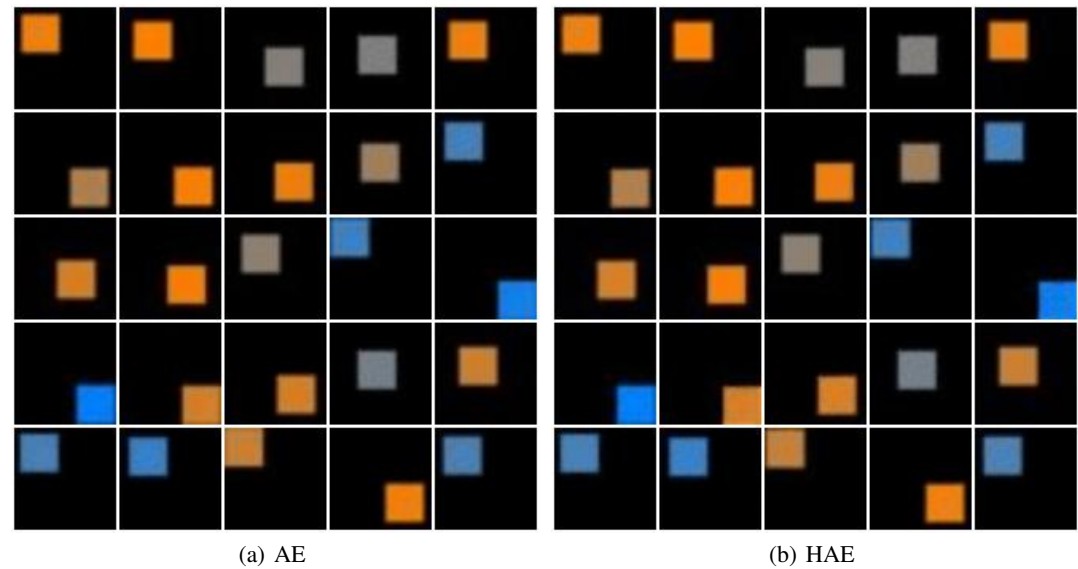

(a) AE                                        (b) HAE

Figure 7: Reconstructions of the AE and HAE models for the `shapes-xc` manifold. The AE was trained for 1000 epochs, and the HAE was trained for 10000 epochs.

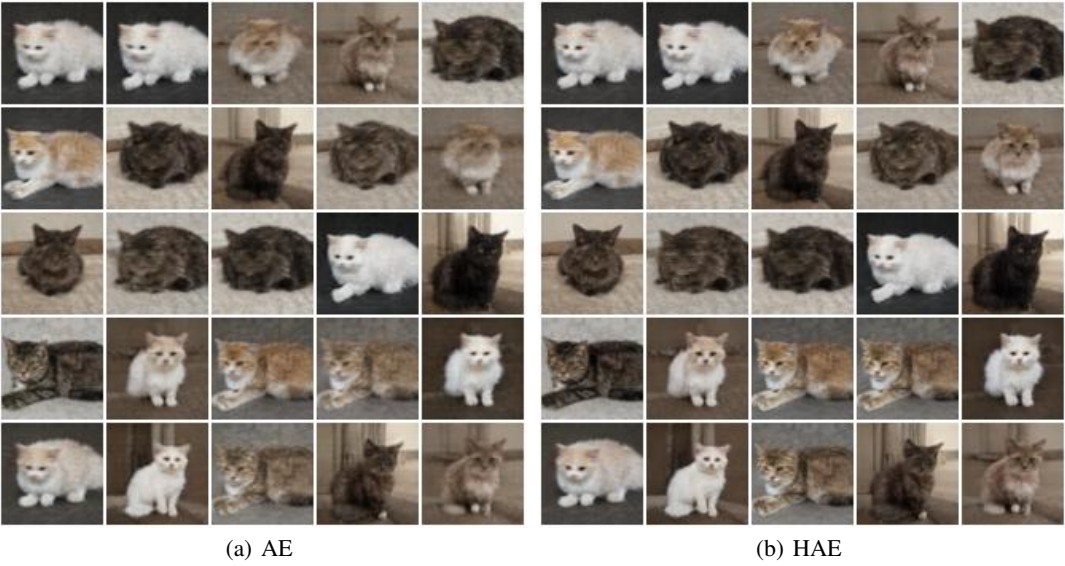

(a) AE                                        (b) HAE

Figure 8: Reconstructions of the AE and HAE models for the `cat1` manifold. The AE was trained for 100 epochs, and the HAE was trained for 1000 epochs.

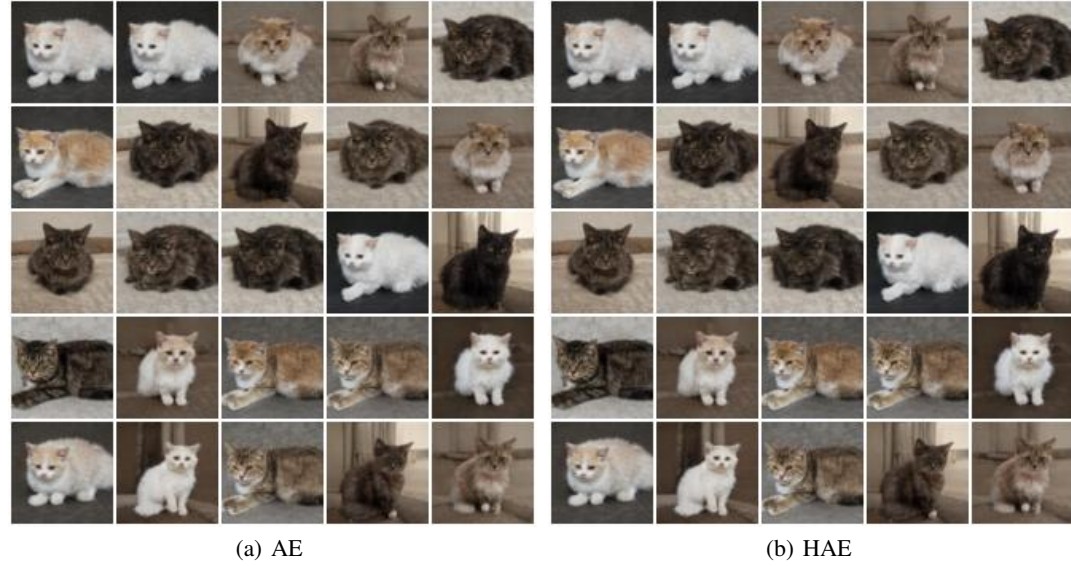

(a) AE                                            (b) HAE

Figure 9: Reconstructions of the AE and HAE models for the `cat1` manifold. The AE was trained for 1000 epochs, and the HAE was trained for 10000 epochs.

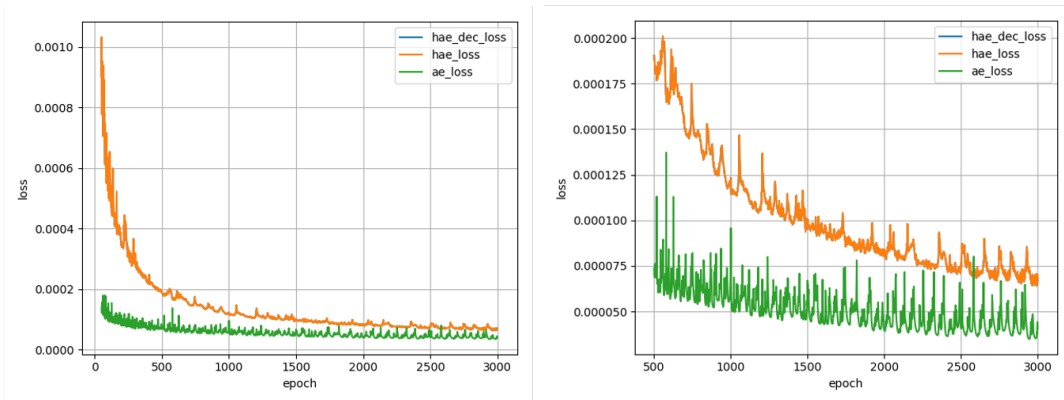

Figure 10:  Training loss of the AE (green) and HAE (orange) models, trained on the `shapes-xc` manifold. The HAE loss is comprised of an HLLE term and a reconstruction term. Thus, we also show the reconstruction term (blue), which cannot be seen as it is essentially identical to the whole HAE loss. The right graph shows a closer examination of the later epochs of training. As can be seen, the gap between the two models decreases with time.

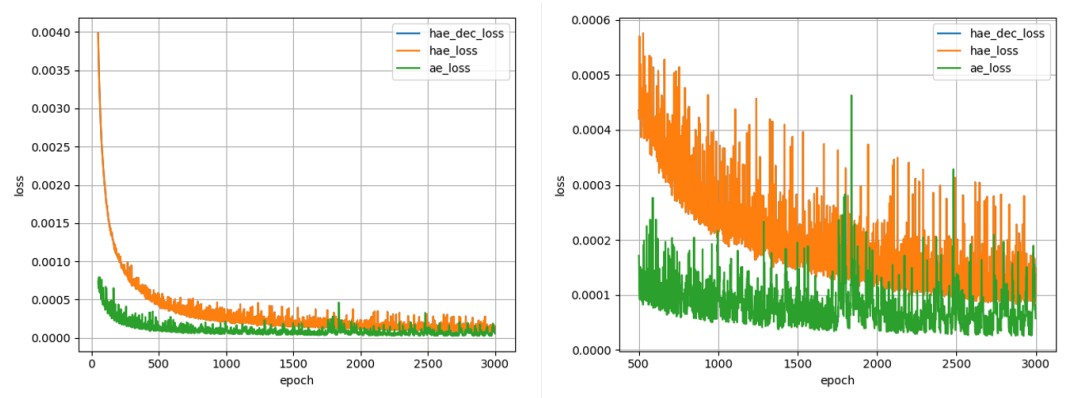

Figure 11: Training loss of the AE (green) and HAE (orange) models, trained on the `cat1` manifold.

**Controlling the manifold.** Sampling specific coordinates of the latent code $z$ of StyleGAN2, results in an entangled manifold. Moreover, our experiments show that latent coordinates have different semantic meanings at different locations in the latent space (see figure 15). In order to define consistent and semantically meaningful latent factors, we follow GANSpace [4] and perform PCA analysis on the $W$ space of StyleGAN2. PCA is performed using $300k$ random $z$ values transformed to $W$ space by the MLP. For each dataset we choose a pair of PCA components with clear and independent semantic meanings (as can be seen in figures 12, 13). We then uniformly sample the two PCA coefficients in the range $[-2\sigma, \ 2\sigma]$ and fix the remaining coefficients at 0.

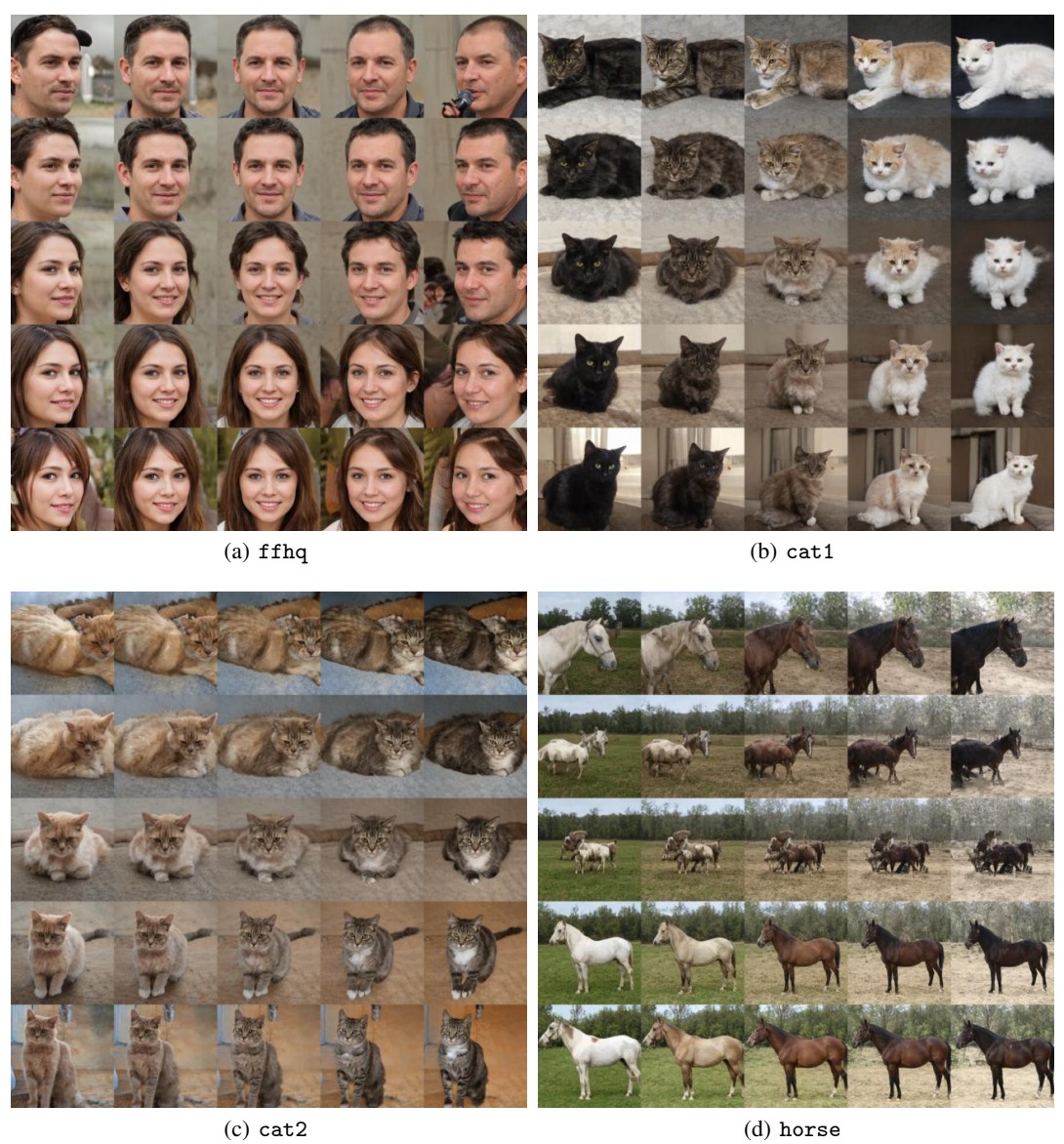

(a) `ffhq`

(b) `cat1`

(c) `cat2`

(d) `horse`

Figure 12: The `ffhq`, `cat1`, `cat2` and `horse` realistic 2D image manifolds, generated with StyleGAN and GANSpace. In each grid, the rows correspond to the $z1$ direction and the columns to the $z2$ direction.

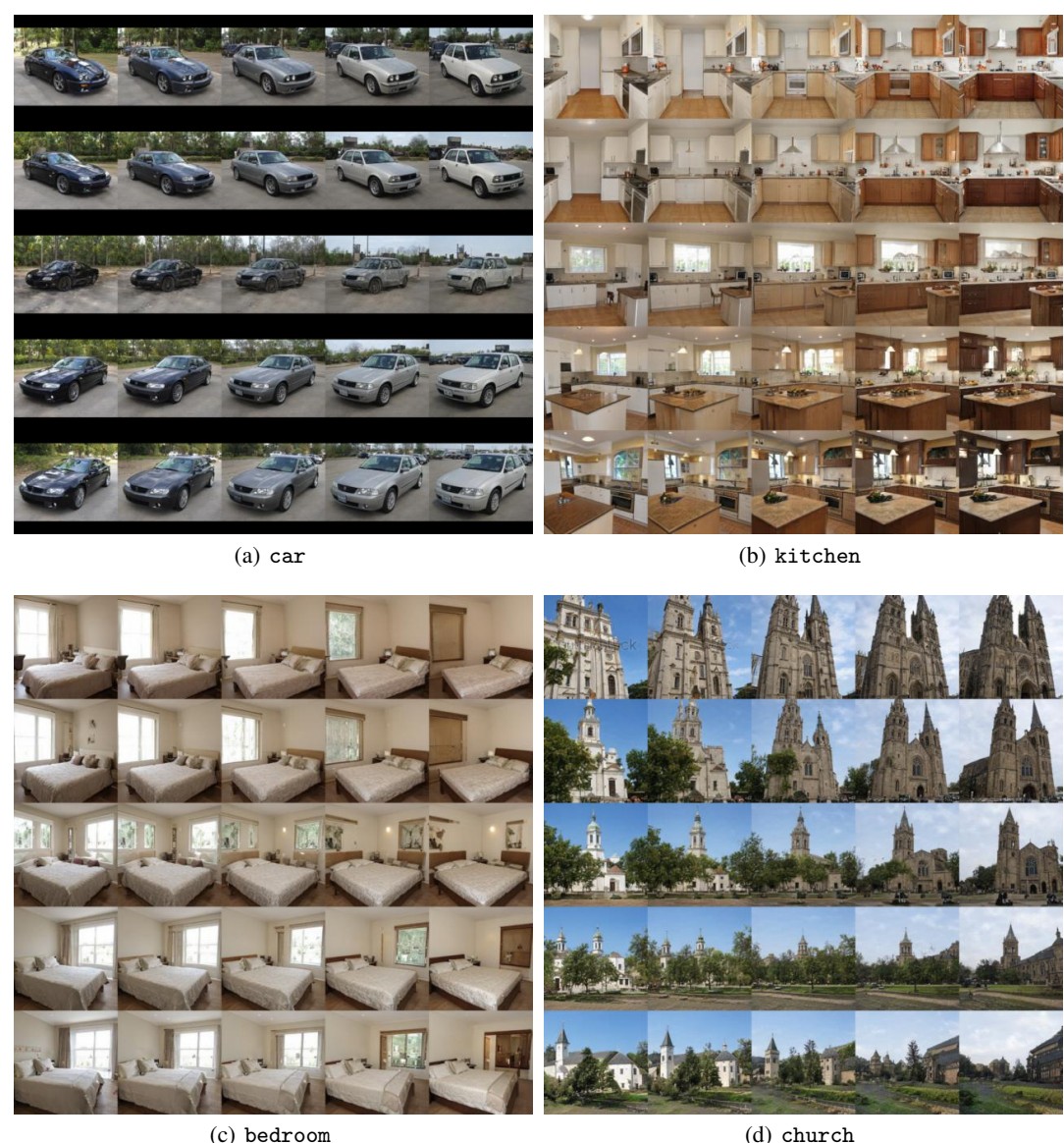

(a) `car`  (b) `kitchen`

(c) `bedroom`  (d) `church`

Figure 13: The `car`, `kitchen`, `bedroom` and `church` realistic 2D image manifolds, generated with StyleGAN and GANSpace. In each grid, the rows correspond to the $z1$ direction and the columns to the $z2$ direction.

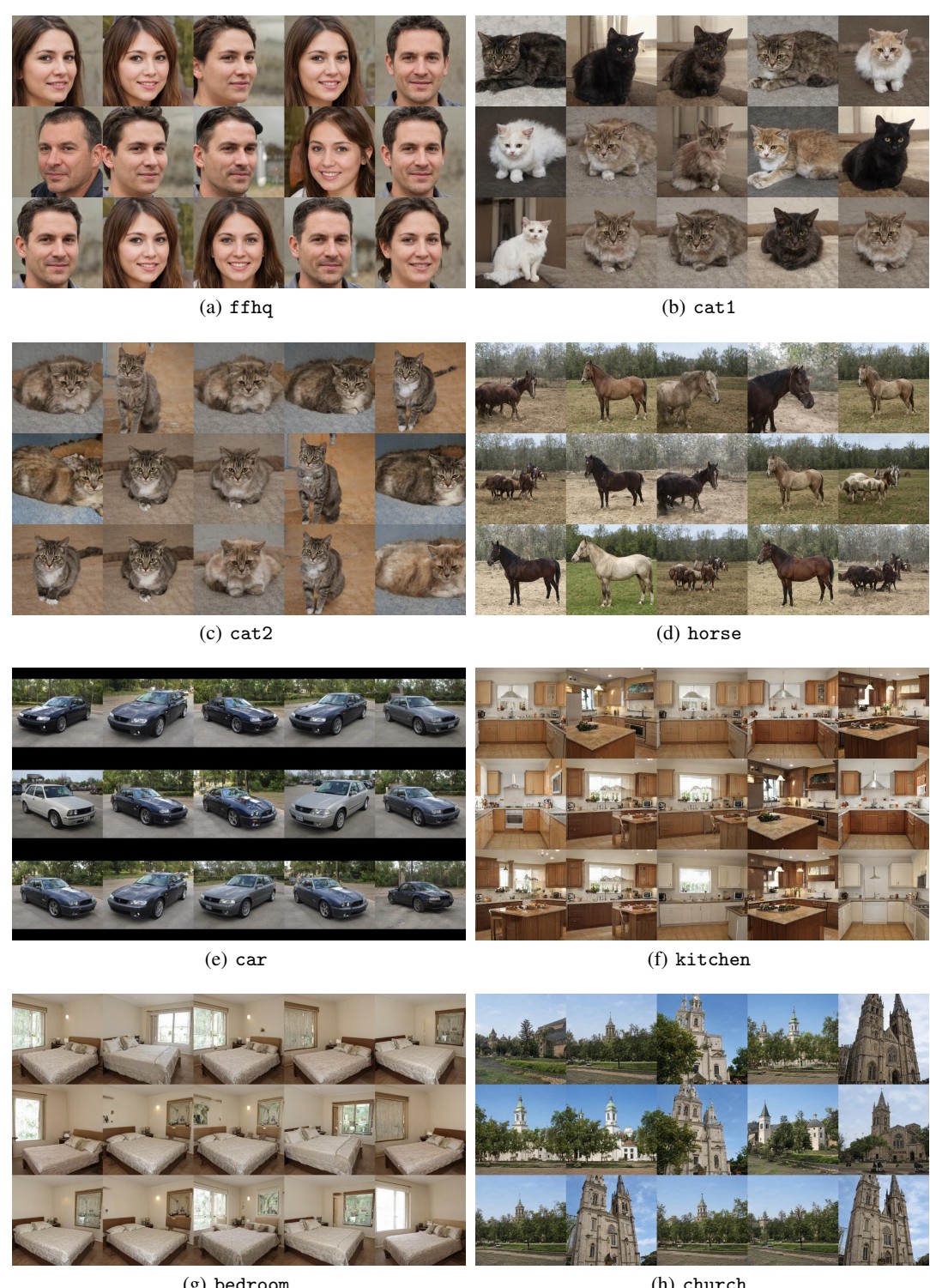

(a) ffhq

(b) cat1

(c) cat2

(d) horse

(e) car

(f) kitchen

(g) bedroom

(h) church

Figure 14: Random samples from the GANSpace 2D manifolds.

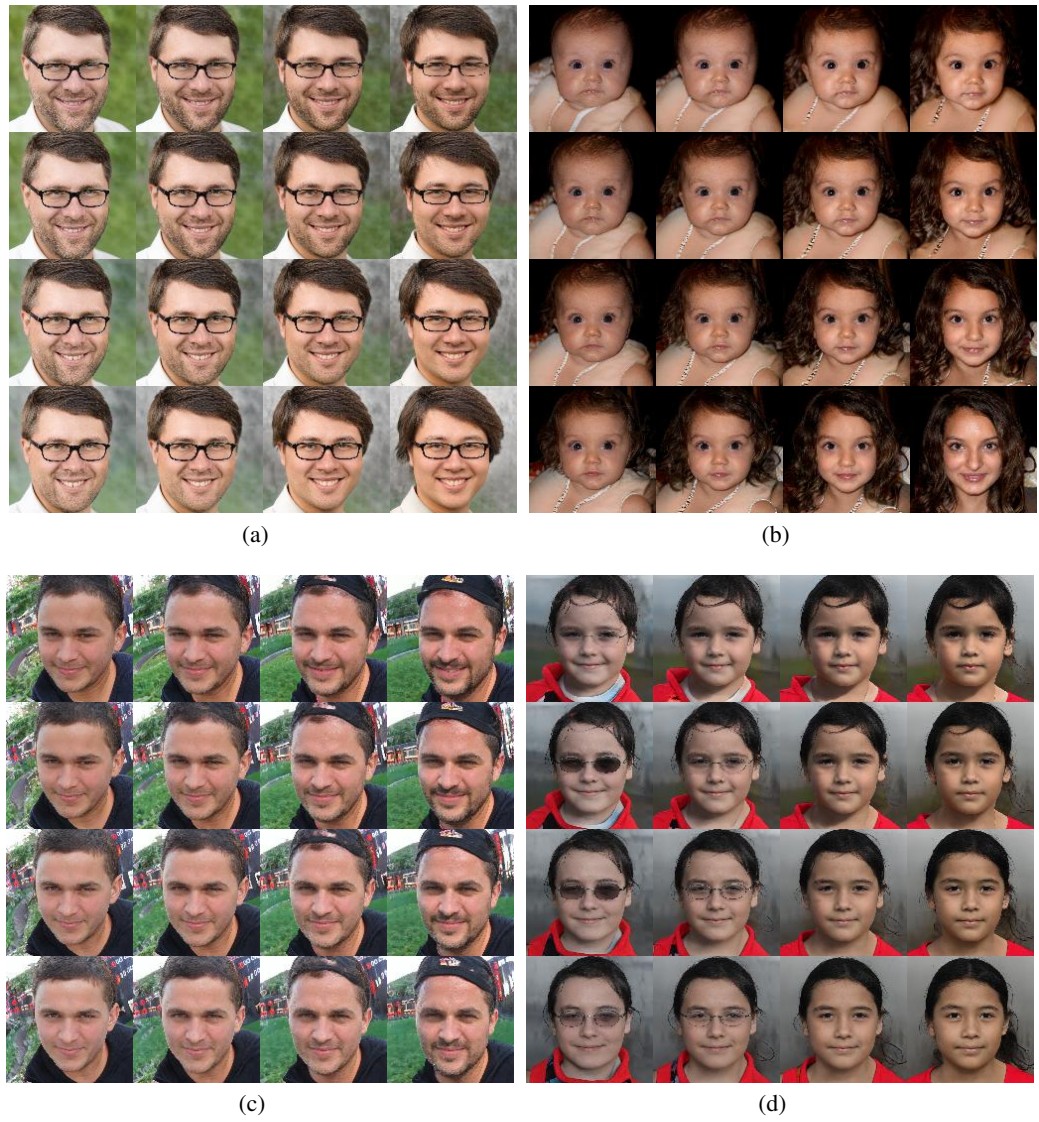

Figure 15: The StyleGAN2 latent codes $z_1$ and $z_2$. In each grid, the rows correspond to $z_1$ and the columns to $z_2$. As can be seen, the two directions have different semantic meanings at different points on the data manifold. Moreover, the manipulation effect is often minor. Thus, we chose to base our manifolds on $W$ directions found using GANSpace, rather than directions in $Z$.