# OpenReview forum: "When Is Unsupervised Disentanglement Possible?"
_NeurIPS.cc/2021/Conference — NeurIPS 2021 Poster_

### Official Review · Reviewer_R2z6 · 2021-07-01

**Rating:** 6
**Confidence:** 4

**Summary:**

The paper follows the recent question of when disentanglement is possible and shows that 'local isometry' is sufficient to ensure that it is possible (not guaranteed) to learn a disentangled representation. This result largely follows from know results in the ICA community alongside the isometry assumption. A proof-of-concepts algorithm that combines HLLE with ICA demonstrates feasibility on synthetic data constructed to satisfy the isometry assumption. While the paper is largely correct, I would argue that it is uninteresting as it (as far as I can tell) only applies to flat manifolds, which is the case we already understand well through linear models.

**Limitations And Societal Impact:**

There is no discussion of societal impact, but this can be forgiven considering the theoretical/conceptual nature of the work.

I do, however, think that the limitations are not sufficiently discussed. As far as I can tell, the assumptions imply that only flat manifolds can be considered, which render the work largely uninteresting (see above).

The final paragraph of the paper has a nice discussion of the limitations of the proposed algorithm and its reliance on HLLE. That was great!

**Main Review:**

The paper considers disentanglement of models where data $x$ is approximated as $x = g(z)$. Here $z$ is a low-dimensional (latent) variable and $g$ is a smooth function. The guiding assumption is that $J^T J = I$, where $J$ is the Jacobian of $g$ and $I$ is the identity matrix. This is assumed true for all $z$. Based on this assumption, the paper shows that classic identifiability results from (Euclidean) ICA holds, such that we can recover the true generative factors (i.e. we can disentangle).

This may sound exciting, but I remain unconvinced. I may have misread the paper, so I am happy to revise my view if the authors can provide more convincing arguments than those put forward in the paper.

Here's my issue:
The matrix $M = J^T J$ ($M$ is a short-hand I here introduce) is the metric tensor of the pull-back metric associated with $g$. If $M = I$ everywhere, then the manifold spanned by $g$ must have zero curvature everywhere, i.e. it must be flat. This leads to two comments from my side:

1. How reasonable is it that actual data manifolds are flat? I would argue that this assumption is so strong that it effectively is uninteresting. Consider data lying on a sphere; even this simple setting is too restrictive for the assumption to work out.

2. In this light, the result that we can disentangle seems rather obvious given that we know that ICA can disentangle linear models. The extension from 'linear' to 'flat' then seem rather incremental. There is still value in the extension, but given that the paper signals that the extension is more significant, I get rather skeptical.

On possible counter-argument to the above is given by the following statement from the paper (lines 97-98):

"
Indeed, assuming that the dimension of z is much smaller than the dimension of x, then any manifold
can be approximated by an isometric mapping. [2].
"

This is, however, all too vague to give this reader much comfort. Clearly any manifold can be approximated by a flat manifold; but can it be well-approximated? Gauss's Theorema Egregium suggest that we perhaps should not be approximating curved objects with flat ones.

On a more positive note, I appreciate that the paper has the relevant proofs as part of the paper rather than hidden in an appendix.  This is great.

I miss a citation to the paper

  Disentangling Disentanglement in Variational Autoencoders
  Emile Mathieu, Tom Rainforth, N. Siddharth, Yee Whye Teh

This gives a partial counter-argument to the Locatello paper, namely that the applied variational inference provides an inductive bias that helps make the models less susceptible to identifiability issues.

# After rebuttal and discussion

I appreciate the rebuttal and follow-up discussions which clarified many things. Based on these I will increase my score to recommend acceptance. **However**, this increase in the score rests on the assumption that the discussions points will be incorporated into the final paper. I think a significant rewrite of several sections of the paper are needed (see our discussions for details) for the paper to be publishable. I trust the authors to implement such changes as it is in their own best interest to do so.

I have increased my score to 6, which I feel is as high as I can go given the state of the submitted manuscript. Had I reviewed a version with the changes implemented, I might have gone higher.


**Time Spent Reviewing:**

2

---

> ### Author Response · Authors · 2021-08-10
> **Response to Reviewer R2z6**
>
> We thank you for your thoughtful review, and address the comments below.
>
> We can certainly understand the concern about “flat manifolds” and welcome the opportunity to correct this misconception.
>
> As we write in line 98, any manifold can be approximated by one that satisfies local isometry. As [2] points out this follows from the Nash-Kuiper theorem (1956) which shows that for any manifold of dimension m embedded in $n$ dimensional space ($n \geq m+1$) and defined by the mapping $f$ and for arbitrary epsilon there exists a function $f_\epsilon$ that is an isometry and $|f_{\epsilon}(x)-f(x)|<\epsilon$.
> (see https://www.abelprize.no/c63466/seksjon/vis.html?tid=63467)
>
> Note that this result does not contradict Gauss’s Theorem Egregium which talks about exact, global isometries. Indeed, it is impossible to find a global, exact isometry from the sphere to the plane but you can easily verify that running HLLE on data sampled from a partial sphere will find an embedding that satisfies local isometry almost perfectly (e.g. when we sample 2000 points on a quarter of a 3D sphere, the local distances in the 2D embedding space found by HLLE and those in the original 3D space have a correlation coefficient of 0.96).
>
> In addition to these theoretical results, please note that the manifolds we deal with here are highly nonlinear: if you take a linear combination of two images of a translating square the result will not be on the manifold (for any two samples in our dataset) and the same holds for images of cats in different poses etc. Calling these manifolds “flat” is highly misleading in our opinion, although they are locally flat. Note also that Donoho and Grimes (2000) have given many examples of highly nonlinear manifolds of interest that provably satisfy local isometry and we review these examples in section 3.
>
> We respectfully disagree with the comment that our results seem “rather obvious given that we know that ICA can disentangle linear models.“ Again, the manifolds we deal with are highly nonlinear and hence ICA will fail miserably to recover the true factors. We are not aware of any existing method that can provably find the correct factors in such nonlinear manifolds.
>
> We also believe that the reviewer’s summary of our paper is missing. Our results not only show that disentanglement is possible with assumptions of local isometry plus non-Gaussianity, but also present an algorithm that is guaranteed to recover them in the limit of infinite data.
>
> Obtaining datasets with corresponding ground truth variables that are also realistic is a known challenge in the field. We chose one way of generating such datasets, and we agree that it is not perfect. However, we feel it is somewhat unfair to say that they were “constructed to satisfy the isometry assumption.” You may not believe us, but we actually created the datasets in order to benchmark different disentanglement algorithms, before we had even derived the connection between local isometry and disentanglement.
>
> We agree that the work of Matthieu et al. should be cited and indeed there is some inductive bias given by the use of variational inference but Locatello’s results (and our own experiments) suggest that this bias is not strong enough. In contrast, our theoretical results and experiments show that the local isometry assumption plus non-Gaussianity are sufficient and we believe that our paper provides a much stronger counter-argument to Locatello’s influential work.

---

### Official Review · Reviewer_tdhX · 2021-07-15

**Rating:** 6
**Confidence:** 4

**Summary:**

The authors propose a novel method to learn disentangled representations from data. Their approach uses some inductive biases on the transformation between the latent and observed spaces, as well as on the latent distribution. Namely, they assume the transformation satisfies the assumption of the local isometry, and that the latent factors are (non-jointly) non-gaussian.
The disentangled representations are learned by first applying a spectral transform to the data, followed by a linear ICA unmixing.
Their method is backed by a theoretical analysis, as well as an empirical validation.

**Limitations And Societal Impact:**

The limitations of the proposed method, as well as how it relates to prior work, were well discussed.

**Main Review:**

The authors propose a novel method for learning disentangled representations, which is an important topic in unsupervised learning.
The proposed approach is based on a spectral decomposition followed by a linear ICA step, instead of learning a full generative model.
This gives it a considerable edge in run time (albeit this was only explored in 2-d), but takes away the generative capabilities. This is somehow addressed by the Hessian Auto-Encoder proposed in section 4.1, although it's not clear how it compares to state of the art generative models like VAEs and GANs (which have seen variants developed for disentangled representations).

Overall, the paper is well presented and easy to read. The proposed method is novel, and, as far as I know, hasn't been proposed before.

This being said, I have 2 serious concerns:
1- The dimension of the latent space in the experiments was $d = 2$. The authors didn't report the performance of their approach for a higher latent dimension, which I find alarming. From personal experience, disentangled/identifiable representations are harder to find in higher dimensions.
In addition, their approach has much shorter run time, but the experiments were only conducted for $d = 2$. Does this still hold for a much larger latent dimension, say $d = 500$?

2- Figure 6 would be much more informative if the authors used a box plot of violin plot. More generally, error bars are missing from all figures and numbers, and there is no indication that the MCC figures reported were averaged over multiple runs and not for a specific seed.

As well as some other comments:
3- It would benefit the presentation if the spectral methods were better explained in the paper, and not simply relegated to the appendix, as they are core to the proposed method. The same can be done for fastICA, without going into the detail of the algorithm, but just explaining what ICA does for e.g.

4- Did the authors explore the impact of applying pointwise nonlinearities to the latent space of the datasets and see how it impacts performance? This would give experimental validation to Theorem 2.

5- The authors should specify that the "dimension of the manifold" is equal to that of the latent space.

6- Terrible notations: it seems that the indices are used to both designate the index of an observation in a dataset (e.g. line 25 $\{x_i\}$) and the component of a multivariate r.v. (e.g. line 39 $y_1(z_1, z_2)$). This is very confusing when referring to the distances between points (e.g. line 66 and 205, $d(x_i, x_j)$).

Overall,  I think that the theoretical results of this paper are worthwhile, but the experimental section lacks depth and polish. I will give a score of 5, but will be happy to increase it if the other reviewers or the authors' rebuttal convinces me to do so.


**Time Spent Reviewing:**

7.5 hours

---

> ### Author Response · Authors · 2021-08-10
> **Response to Reviewer tdhX**
>
> We thank you for your comments, and address them below.
>
> Regarding the experiments with higher dimensions, as we acknowledged in the paper, spectral methods require dense sampling of the manifold and so the higher the intrinsic dimensionality, the more samples are needed. We do not think that this diminishes from the importance of our theoretical result which shows that disentanglement is possible given our two assumptions, regardless of the dimension, but we agree that devising practical methods for taking advantage of these two assumptions with sparse sampling is an excellent direction for future research.  Nevertheless, we have obtained excellent results with some manifolds whose intrinsic dimensionality is around 5 and we will be happy to include these in the next version.
>
> We have not performed any experiments in which the true dimensionality of the manifold is d=500 as you suggest, but we are not sure that this is realistic: we believe that the fundamental assumption behind manifold learning is that the intrinsic dimensionality is small (e.g. the intrinsic dimensionality of faces is often assumed to be 20-30, even though they are embedded in much higher dimensions). It is true that GANs often require hundreds of latent variables to represent realistic manifolds but this is precisely because their latent factors are entangled. Thus the GANspace paper shows that “100 principal components are sufficient to describe overall image
> appearance” in StyleGAN2 and these principal components are still highly redundant so that a truly independent latent representation should have lower dimensionality. At any rate, as we note above, our theoretical results will also hold for high dimensional manifolds provided that they are sampled densely enough.
>
> We agree we should have used a box plot or violin plot for figure 6, and will definitely do so in the next version. We can assure you that the results are not cherry-picked, and the HAE model has much higher mean, median, min and max values, and much lower std, between the relevant methods. In addition, it is important to note that HLLE does not depend on a random seed, and since HAE converges to almost identical embeddings to HLLE_ICA (as can be seen in figure 1 and 5 e.g.), HAE is not dependent on this randomness either. Regarding the vanilla autoencoders, the results did not change significantly across multiple runs.
>
> We have actually explored the effect of pointwise nonlinearities. This can be seen in figure 2, where the latent factors are a scaled representation of the locally isometric representation, and thus HLLE succeeds even though LEM fails. We have also experimented more directly with applying pointwise nonlinearities to the independent latent factors and observed that indeed HLLE continues to succeed while LEM fails, as expected from theorems 1 and 2.  We will be glad to include these additional results in the next version. We will also fix the notation as you suggest and include more details about the spectral methods and their run times. Note that the run times of spectral methods are dominated by the number of datapoints (since they require creating an N by N matrix and finding its eigendecomposition) and in our implementation there is no dependence of the latent dimension on the running times (aside from our comment regarding the need for dense sampling), hence the “edge in run time” will hold for high dimensional manifolds as well.

---

> > ### Comment · Reviewer_tdhX · 2021-08-25
> > **Thank you and score update**
> >
> > Thank you for the great rebuttal, and for providing more details about local isometry.
> >
> > As I said in my review, I think that the theoretical contribution is novel and important for the representation learning community. I just felt that something was missing in the narrative, with the description of the spectral methods being relegated to the appendix, and the significance of local isometry being brushed over.
> >
> > I have to admit that I was quick to judge the experiments being mostly done in 2D as a serious limitation. I agree that the contribution of this paper is mainly theoretical, and it's nice to see experiments positively supporting the theoretical claims.
> >
> > After reading the rebuttal and the discussion with the other reviewers, I'm happy to raise my score to help the paper getting accepted.

---

### Official Review · Reviewer_pb6M · 2021-07-16

**Rating:** 4
**Confidence:** 4

**Summary:**

This paper studies the disentanglement of samples generated by GANs. It argues that the local isometry and non-Gaussian property are the keys to disentanglement. Some intuitive experiments based on GAN latent space discovery are conducted to support this insight.

**Limitations And Societal Impact:**

The paper mentioned the limitations in Sec.6.

**Main Review:**

Pros:
1. The writing is clear and easy to follow.
2. The review of the traditional manifold learning method is helpful for current GAN research.

Concerns:
1. The two theorems in this paper are a bit tricky. First, a local isometry is a very strong condition, which basically cannot happen in practice. Second, these two theorems do not explain why intuitively disentanglement is connected with isometry, and what is the role of non-Gaussian property? Third, is there any in-depth analysis on the disentangled representations? For example, what is the convergence rate to the disentangled representations with respect to the data volume? Are there any more efficient algorithms? Is isometry or non-Gaussian necessary in disentanglement?
2. I must correct a logical mistake in this paper. That "you can get disentangled representations at local isometry and non-Gaussian condition" does not mean "you cannot get disentangled representations otherwise." So the content of this paper has presented cannot answer the question 'is disentanglement possible.'
3. The experiments are too weak. Why is GANSpace sufficient to support your claim? Any insights or theory?

**Time Spent Reviewing:**

5

---

> ### Author Response · Authors · 2021-08-10
> **Response to Reviewer pb6M**
>
> We thank you for your comments, and address them below.
>
> We respectfully disagree with the comment that “local isometry is a strong condition which basically cannot happen in practice”.  As we show in section 3, there are many manifolds used in practice that can be shown to provably satisfy this condition. Furthermore, as we write in line 98, any manifold can be approximated by one that satisfies local isometry. As [2] points out this follows from the Nash-Kuiper theorem (1956) and we will briefly review this classical result in the next version.
>
> The role of non-Gaussianity is crucial to solve a linear ambiguity as we discuss in line 54 and is well-known from the ICA literature. If the latent factors have a Gaussian distribution, then any linear mixing of them will also have a Gaussian distribution. Thus local isometry allows us to solve disentanglement up to a linear mixing and non-Gaussianity allows us to solve the linear mixing (see figure 3, left).
>
> Regarding the “logical mistake”, we wish to note that Locatello et al. have already shown that disentanglement is impossible without prior assumptions. Our paper shows that two assumptions are sufficient and these two assumptions are satisfied in many interesting, realistic cases. We agree that we have not shown that these assumptions are necessary but we believe this is indeed reflected in how we wrote the paper (e.g. the caption of figure 1).
>
> Regarding the experiments, our main result is theoretical and does not rest on GANspace in any way. The role of the experiments is to show that the theoretical result (which assumes perfect isometry and infinite data) also holds for finite datasets and for realistic manifolds.

---

> > ### Comment · Reviewer_pb6M · 2021-08-29
> > **Discuss**
> >
> > First, sorry about the later reply.
> >
> >
> > I’m very happy to receive the authors’ feedback. It does clarify some aspects of this paper and improves my overall impression of it. However, let me explain my major concern about local isometry.
> >
> >
> > I know that this has been widely used in traditional manifold learning literature. Still, for me, it is also the major reason that manifold learning has no longer been that popular in the recent 10 years. Overall, the performance of those traditional manifold learning methods with such strong regularity assumptions cannot compete with deep learning methods. Those deep learning models are often proved to lack good regularity properties. However, this is just my own opinion, so I will not take it into judging this paper.
> >
> >
> > What really concerns me is that in this paper, we are researching the property of the generated manifold, not the real data manifold. The generated manifold is induced by the generator, which is an explicit function that we know its analytic representation. So two points come after it: even if the data manifold holds local isometry to some Euclidean space, the generated manifold may not also hold it, as they are different (we can directly measure the Jensen Shannon divergence or estimate the Wasserstein divergence to see they are different, and there are some theoretical works that also proves they are different.); we can directly compute the Jacobian of the generator function to see whether it is a local isometry, that is, whether $J ^TJ =I$. I have personally checked it, and the authors may also check it, and I find that apparently, it is not local isometry.
> >
> > The authors may argue that the above computation is done in ambient space, which cannot reflect the property of real data manifold. But let me emphasize again that we are studying disentanglement of the generated manifold, not the real data manifold, at least in this paper.
> >
> > In fact, GANs are known for mode collapse and mode drop, which suggests two points: the generated manifold is not the real data manifold, and their distance cannot be ignored; the Jacobian of the generator does not hold good regularity property. For the real Jacobian of the generators regarding the manifold, some previous theory work [1] proves that they are very easy to vanish or explode. So theoretically, they are also very hard to be local isometry.
> >
> > In fact, if you look into the first eight MLP layers of StyleGAN, you will find it totally ill-conditioned, which means even the very shadow $W$ space is completely not local isometry to the $Z$ space.
> >
> > For the time being, many GAN models use the regularization technique to the Jacobian of its generator. Thus I am highly suspect of the generality of the theory in this paper.
> >
> >
> > Hence, please let me insist on my original score. I will be pleased if the authors or other reviewers can correct me.
> >
> > 1. Towards Principled Methods for Training Generative Adversarial Networks.

---

> > > ### Author Response · Authors · 2021-08-29
> > > **Important clarification**
> > >
> > > Thank you for your comments, it led us to realize what may have been misunderstood in the first place. We will try to use the opportunity to clarify this.
> > >
> > > Our goal in the paper is to ask: is disentanglement possible?
> > >
> > > By disentanglement we (and most people in the community) mean recovering the "true" factors. For example in the case of faces, these "true" factors are things like expression, pose, illumination etc. In the case of simple geometric figures such as dsprites, the true factors are things like location, color and orientation. These true factors are the factors we analyze in the paper, and for which local isometry needs to be satisfied, for the theorem to hold.
> > >
> > > We do NOT mean recovering the latent z's of a GAN. This is for precisely the reason that you wrote in your review: the latent z's of a GAN may be very far from the desired true factors and thus may themselves be highly entangled. This is also why we used GANspace and not the original GAN z's in the experiments: the authors of GANspace have shown that the top PCA coefficients of the W space are often correlated with the true factors, even though the latent z's are not.
> > >
> > > Thus the question that is examined is not whether the generator function of a GAN satisfies local isometry, but whether the mapping from the ground truth factors to the data satisfies it (up to a linear transformation and pointwise nonlinearity). As Donoho and Grimes have shown, the answer to this question is positive for many interesting manifolds, and our experiments support the theoretical results.
> > >
> > > We hope this clarifies this misunderstanding, and would be happy to answer any additional questions.

---

> > > > ### Comment · Reviewer_pb6M · 2021-08-31
> > > > **Discussion**
> > > >
> > > > About "Thus the question that is examined is not whether the generator function of a GAN satisfies local isometry, but whether **the mapping from the ground truth factors to the data satisfies it**". What's this kind of mapping look like? Does it exist in the real world? And the authors used GANSpace and Hessian Auto-Encoder in the experiment. Does that mean that the GANs and Auto-Encoder belong to those kinds of mapping?
> > > >
> > > > By the way, in your first response to me *Why is GANSpace sufficient to support your claim? Any insights or theory?* Your answer is *Regarding the experiments, our main result is theoretical and does not rest on GANspace in any way.*  But in the latter response, you said *This is also why we used GANSpace...*

---

> > > > > ### Author Response · Authors · 2021-09-01
> > > > > **Response to Reviewer pb6M**
> > > > >
> > > > > Our paper is divided into a theoretical section and an experiments section. The former is the more significant part; the latter is there to support it.
> > > > >
> > > > > In the theoretical part, we prove (theoretically, without considering any specific manifold or dataset) that disentanglement is possible, if the assumptions (namely, local isometry and non gaussianity) hold. In this part, we do not rest on GANspace or any other GAN, network or model for that matter. This theorem is true for any data of interest, as long as it satisfies the assumptions.
> > > > >
> > > > > In the experiments, we aim to show that these assumptions hold in practice. First we show that there exist many different manifolds that indeed satisfy the assumptions (referencing Donoho and Grimes). Then, we generate datasets of two kinds: (1) very simple synthetic datasets of squares, and (2) more realistic manifolds, using GANspace. The results support the theory. We see that the assumptions hold in in these experiments, and thus that disentanglement is indeed possible on these generated manifolds.
> > > > >
> > > > > So our main result, the theoretical one, does not rest on GANspace in any way. In some of our experiments, indeed we used GANspace, and we explained why we chose it over alternative models.
> > > > >
> > > > > Regarding your question, the mapping we refer to is the true mapping function that is assumed for real world data under the manifold hypothesis, and to which we are blind to.

---

### Official Review · Reviewer_nrJU · 2021-07-16

**Rating:** 7
**Confidence:** 3

**Summary:**

The paper shows that having local isometry together with non-Gaussianity of latent variables and sufficient number of samples guaranteed to learn a disentangled representation, without any domain-specific knowledge. The authors discussed some potential manifolds for which local isometry is satisfied. The authors also conducted several experiments to evaluate the disentanglement of the obtained representation by their method in contrast to the comparable approaches. The authors demonstrated the latent embedding and the mean correlation coefficient as qualitative and quantitative disentanglement measures, respectively.



**Limitations And Societal Impact:**

- As discussed by the authors, there are some practical limitations for using the HILLE+ICA in representation learning. ICA/fastICA suffers from convergence issue and it is not easy to substitute ICA with another method to satisfy non-Gaussianity.

- According to Theorem 1, latent dimensions are assumed to be uniform, which is a strong assumption in many domains of application. For instance, in the gene expression dataset which is also mentioned in section 3, such uniformity is not present. Can the authors explain how the theoretical results are satisfied for dataset with heterogeneity and rare observations?

- It seems that both HILLE and HAE suffer from poorer reconstruction compared to AE. Is this due to the objective in Eq.1 ?


**Main Review:**

Originality:
The paper is addressing a challenging and important problem. Although the local isometry property has been studied before, the authors provide a new theoretical analysis for the disentangled representation which is quite valuable and expands the finding of the previous disentanglement studies by Locatello et al., 2018, 2020. They show that in the presence of enough training samples, satisfying two conditions: isometry and non-Gaussianity on the latent space is enough to obtain disentangled factors. They also discuss the sensitivity of the isometry to the parametrization of each factor.

Quality:
The paper is technically sound with enough theory and experimental studies. Theorem 1 shows if there exist an isometric mapping from latent space to the data space, using Hessian eigenmaps and ICA, the latent representation is disentangled. Theorem 2 generalizes the result of the first theorem by considering another mapping from latent space to an alternative space, which relaxes the local isometry constraint on the latent factors.

Clarity:
The paper is well written and well organized.

Significance:
Both theoretical and experimental results seem valuable. Experimental results show that both HILLE+ICA and HAE obtain significantly better MCC compared to the prior methods. The results show that the HILLE+ICA and HAE have roughly the same reconstruction, generation, and interpolation performance compared to the vanilla AE.


**Time Spent Reviewing:**

12

---

> ### Author Response · Authors · 2021-08-10
> **Response to Reviewer nrJU**
>
> We thank you for your thoughtful review, and address the comments below.
>
> In Theorem 1 we assumed the latent factors had a uniform distribution but note that in Theorem 2 we no longer require it. We required it in Theorem 1 to use classical results on Laplacian EigenMaps but this requirement is not needed for Hessian EigenMaps which is what we use in the experiments. Theorem 2 only requires that the source distribution be sufficiently non-Gaussian for ICA algorithms to work, so it can accommodate rare observations and heterogeneity.
>
> The HAE reconstructions are indeed poorer than the vanilla auto-encoder for a small number of iterations, since the loss includes a second term in addition to reconstruction. However, we found that when more iterations are used, both terms in the loss are minimized well and the reconstructions with more iterations are of the same high quality as the vanilla auto-encoder. We will include these results in the next version.
>
> Regarding the final ICA step, we agree that there may be convergence issues with some datasets. Note however that for some uses of representation learning the ICA step is not needed since it only performs a linear unmixing of the representation. For example, if a linear classifier is trained on the representation (as is often done), then ICA is not needed (since the classifier can learn the linear transformation).

---

> > ### Comment · Reviewer_nrJU · 2021-08-22
> > **Response to the rebuttal**
> >
> > Thanks for the response. The authors addressed some of my concerns. I still have some questions and suggestions.
> >
> > - Theorem 2: The authors mentioned that in Theorem 2, there is no need for uniform distribution assumption and the isometry will be preserved even in the presence of non-uniform (imbalanced) distribution. However, I do not see how the proposed method locally preserves the distance in an imbalanced case with rare observations, when in theory the infinite number of samples are required. Would you please further clarify how your model handles a non-uniform case?
> > - “we found that when more iterations are used, both terms in the loss are minimized well and the reconstructions with more iterations are of the same high quality as the vanilla auto-encoder”: I did not expect this, since you are imposing two strong constraints on a very low-dimensional manifold (2D space). Therefore, the poorer reconstruction is only due to the slower convergence, correct?
> >   I highly suggest including your new results in the manuscript.
> > - “there may be convergence issues with some datasets”: I recommend the authors to further explain the limitations of their method in sec. 6 and address the above discussion.

---

> > > ### Author Response · Authors · 2021-08-25
> > > **Response to Reviewer nrJU's response**
> > >
> > > Thank you for your comments.
> > >
> > > Regarding rare observations, the non-Gaussianity requirements of our method are essentially those of ICA methods. There is actually quite a bit of experience in applying these to non-uniform data, for example:
> > > https://scikit-learn.org/stable/auto_examples/decomposition/plot_ica_vs_pca.html#sphx-glr-auto-examples-decomposition-plot-ica-vs-pca-py
> > >
> > > Yes, you are correct. We found that the poorer reconstructions are only due to slower convergence. When we allowed for longer training, the Hessian AutoEncoder was able to achieve the same high resolution results as the vanilla model.
> > >
> > > We will be sure to include these results and the discussion you mentioned in the next version, as you suggested.

---

### Author Response · Authors · 2021-08-18
**Local Isometry clarification**

Dear Reviewers,

The area chair has asked us to provide more detail and intuition about why the local isometry assumption is reasonable at the global scale needed for disentanglement.
We are happy to do so, and will gladly include this background material in the next version of the paper.

First, we should emphasize that local isometry is not an assumption that we invented for the sake of this paper, but rather is widely used in the classical manifold learning literature. Briefly, Saul and Roweis (2000) used the assumption that manifolds are locally linear as the basis of LLE, and Tenenbaum (2000) used the stronger assumption of global isometry as the basis of ISOMAP. Donoho and Grimes later (2003) showed how to interpret LLE in terms of local isometry and presented HLLE as a more principled alternative to LLE. They also showed that local isometry is a more general assumption than the global one used by ISOMAP, and which covers many interesting articulation manifolds. Taken together, these algorithms have been used in a remarkably wide range of applications of manifold learning, and the papers have been cited over 33 thousand times.

Manifold learning approaches based on local isometry have been shown to be successful on precisely the type of manifolds that are typically used in the disentanglement literature. Many recent papers use dsprites, or some variant of it, which consists of simple 2D objects translating and rotating in the image plane. These types of manifolds were shown to satisfy local isometry by Donoho and Grimes (2005) (see lines 160-163 of our paper). An additional example that is very common in the disentanglement literature is images of faces at different poses, illumination and expressions. Donoho and Grimes (2004) have shown that the manifold of images of faces with different expressions satisfies local isometry, and such datasets are successfully handled with HLLE.

Finally, we hope to give some intuition about why scale is not a problem.
Consider a partial circle, which is a 1D manifold in 2D. As discussed in the paper, as long as we use the parameterization $(cos(z),sin(z))$, then an $\epsilon$ change in $z$ will correspond to an $\epsilon$ change in $x$. Only when we use the full circle do we encounter a problem, in which case a small change in $x$ can correspond to a large change in $z$ (from $0$ to $2\pi$) (this was the reason we used a partial sphere and not the full sphere in our rebuttal). However, the important thing to note here is that the problem is a result of a “wrap around”, and not due to the large scale.
Another useful intuition is that of “arc-length parameterization.” Any curve (1D manifold) in high dimensions $x(t)$ can be reparameterized in terms of arc-length $x(s)$. When we use the arc-length parameterization, then $J^TJ=I$ everywhere and local isometry is satisfied. The Nash-Kuiper theorem shows that this intuition generalizes to higher-dimensional manifolds as well (to arbitrary accuracy).

We hope this helps clarify any ambiguity, and will be happy to provide any additional clarification.

---

### Decision · Program_Chairs · 2021-09-27

**Decision:**

Accept (Poster)

**Comment:**

This paper examines the hotly debated topic of whether one can learn disentangled representations from unlabelled data.  The key contribution is to show that a combination of non-Gaussianity and local isometry is sufficient to make unsupervised disentanglement theoretically possible.  This is then backed up by a proof-of-concept algorithm and experiments.

After receiving initially quite mixed review scores, this paper has undergone substantial discussions by the reviewers, both publically with the authors and behind closed doors, resulting in two reviewers who initially recommend rejection increasing their scores to back marginal acceptance.  Perhaps the key underlying issue that has been debated is how strong the assumptions required by the theoretical results actually are in practice.  After much back and forth, the general consensus on this seems to be that the assumptions are actually not unreasonable, but need to be much more carefully explained and discussed in the paper.

More generally, the overall sentiment of the reviewers, which I concur with myself, is that the underlying ideas in the paper are interesting and potentially quite significant, but that noticeable work on the writing of the paper itself is needed to improve the clarity and ensure the work can be properly appreciated by its audience.  Though, at the time of writing, Reviewer pb6M still also has concerns about the relationship between the theory and experiments in the paper and which mapping the local isometry assumption applies to, I agree with the authors that this concern seems to be based on a misunderstanding rather than a genuine issue.

Putting this together, I believe that the decision of acceptance comes mostly down to whether the clarity issues are deemed too severe for publication / require an unacceptably large amount of changes for the camera-ready.  I think this is quite a tight call, but recommend giving the author's the benefit of the doubt that the required changes will be adequately incorporated.  The basis for this is that I think that the work does have the potential to stimulate notable interest in an area that is quickly moving, such that the risk of the clarity issues not being solved is outweighed by the potential advantages of ensuring the ideas are published.

If this decision is upheld, I sincerely hope that the authors will make the required efforts to improve the paper for the camera-ready as the changes required are non-trivial and it would be a shame to leave the writing of the paper letting down the underlying technical content.  I would also very strongly suggest the authors change the title of the paper, which is inappropriately general and does a disservice to previous work that has considered the same question.